# ICDiffAD: Implicit Conditioning Diffusion Model for Time Series Anomaly Detection

**Fan Zhang**[1]**, Sinchee Chin**[1]**, Jing-Hao Xue**[2]**, Wenming Yang**[1][†]

[1] Shenzhen International Graduate School, Tsinghua University, Shenzhen, China

[2] Department of Statistical Science, University College London, London, United Kingdom
`{zf23, chenxz22}@mails.tsinghua.edu.cn, jinghao.xue@ucl.ac.uk,`
`yang.wenming@sz.tsinghua.edu.cn`

## Abstract

Time series anomaly detection (TSAD) faces critical challenges from intrinsic data noisiness and temporal heterogeneity, which undermine the reconstruction fidelity of prevailing generative approaches. While diffusion models offer theoretical advantages in capturing complex temporal dynamics, their inherent stochasticity introduces irreducible variance in reconstructions. We present ICDiffAD, a novel method that synergizes adaptive noise scheduling with input-consistent generation to address these limitations. ICDiffAD introduces two key innovations: (1) an **SNR Scheduler** that governs training through quantifiable noise scales, enabling robust learning of normative patterns across non-stationary regimes; and (2) an **SNR Implicit Conditioning Mechanism** that initializes reverse diffusion from partially corrupted inputs, preserving signal coherence while attenuating anomalous components. ICDiffAD ensures high-fidelity reconstructions aligned with the input's manifold, reconciling generative flexibility with detection accuracy. Across five multivariate benchmarks, ICDiffAD improves the F1 score by 19.57% and reduces false positives by 60.23% compared to existing diffusion model-based TSAD methods.

## 1 Introduction

Time series anomaly detection (TSAD) is a key component of many essential applications, such as maintenance and surveillance (Dissanayake et al., 2023; Nguyen-Da et al., 2024). However, TSAD is faced with two challenges: (i) **intrinsic noisiness of temporal data streams**, where anomalous signals are obfuscated by stochastic fluctuations, sensor artefacts, and measurement inaccuracies; and (ii) **temporal heterogeneity of real-world time series**, characterised by non-stationarity and regime shifts, which obstruct the learning of robust latent patterns (Liu et al., 2021; Wang and Yin, 2024).

Among prevailing approaches, reconstruction-based methods operationalizing anomaly detection through normative dynamics approximation dominate contemporary TSAD research. These methods leverage deep learning models such as autoencoders, variational autoencoders, and Transformer-based sequence models to learn the underlying premise posits of the anomaly-free training samples. However, the practical efficacy of these approaches is severely impeded by the inherent noise and high temporal heterogeneity of time series data. The presence of random background noise in the training data compromises the model's capacity to accurately capture the underlying dynamics of normal system behavior, resulting in a model memorizing noise artifacts rather than learning causal temporal dependencies (Hundman et al., 2018; Saravanan et al., 2023). Simultaneously, the high temporal heterogeneity inherent to real-world time series manifests as non-stationary distributions and regime shifts, destabilizing training dynamics (Zhan et al., 2024). These two limitations fundamentally constrain the reconstruction fidelity, thereby increasing false positives in TSAD.

Recently, researchers have focused on the use of generative models such as GANs and diffusion models for TSAD, motivated by their ability to asymptotically approximate complex data manifolds through iterative denoising (Goodfellow et al., 2020; Ho et al., 2020). Unlike deterministic

---

[†]Corresponding author.

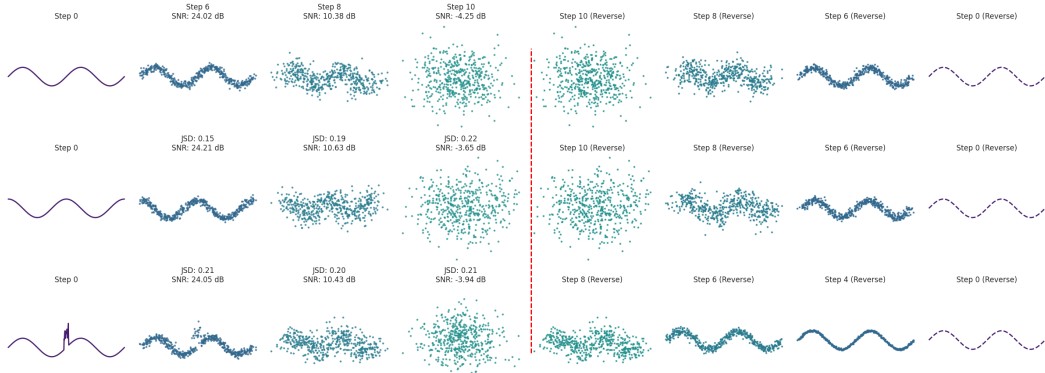

Figure 1: Diffusion dynamics for Sinusoidal waveforms. (Top) Sine wave. (Middle) Cosine wave. (Bottom) Sine wave with injected anomaly. The forward diffusion process progressively attenuates anomalous signals; the reverse process reconstructs the original time series. Performance metrics (SNR and Jensen-Shannon Divergence (JSD)) are measured relative to step 0. At SNR $\simeq -4$ dB, the original signal's trends are fully attenuated, leading to stochastic reconstruction (e.g., sine waves reconstructed as cosine waves), degrading TSAD performance. In contrast, at an optimized denoising step (SNR $\simeq 10.5$ dB), anomalous signals are suppressed while preserving the underlying trend, enabling faithful reconstruction and improved TSAD accuracy.

autoencoders, diffusion models learn to progressively refine a noise-corrupted sample toward the data distribution, theoretically circumventing the over-constrained reconstruction inherent to traditional methods. However, a critical issue remains under-explored: generative models inherently generate **diverse plausible reconstructions** from Gaussian noise (e.g., as shown in Figure 1, reconstructing sine waves as cosine waves at low Signal-to-Noise Ratio (SNR)), contradicting reconstruction-based TSAD's need for input-consistent reconstructions and degrading TSAD performance. This stochasticity significantly increases false positive rates, especially in high-variance regions where multiple reconstructions may seem equally likely. Although some studies attempt to mitigate this by introducing self-conditioning or partial interpolation, they risk conditioning on anomalous points (Xiao et al., 2023; Chen et al., 2023), which undermines temporal consistency and leads to biased reconstructions where local anomalies may be mistakenly preserved.

These limitations motivate our investigation on implicit-conditioning for input-consistent generation to revamp TSAD. Hence in this paper we propose **ICDiffAD**, a new and effective TSAD paradigm equipped with two exciting innovations: (1) **SNR Scheduler**, a reparameterized training-inference scheme that unifies noise scales under a quantifiable SNR spectrum. Unlike conventional fixed or linear schedules, this new scheduler induces the model to learn normal patterns across diverse, SNR-calibrated corruption levels, enhancing robustness to noise perturbations and temporal heterogeneity (§3.2); and (2) **SNR Implicit Conditioning (SIC) Mechanism**, a corruption estimator that jointly predicts (i) the optimal corruption factor $\bar{\alpha}_{\hat{T}}$ and (ii) the denoising step $\hat{T}$ for a given input. By dynamically aligning the reverse process with the input's intrinsic noise scale, SIC ensures that reconstructions balance **faithfulness** (avoiding unrealistic outputs) and **fidelity** (preserving non-anomalous trends), resolving the realism-fidelity trade-off inherent to vanilla diffusion in TSAD.

Notably, ICDiffAD's design explicitly addresses the TSAD-specific need for reconstruction determinism: by conditioning generation on SNR-inferred latent states, we suppress stochastic deviations while attenuating anomalies. As shown in §5, ICDiffAD provides significant improvement over previous diffusion models (+20.2% average F1 score) and reconstruction-based methods (+5.1% average F1 score) in TSAD. In short, our contributions are threefold.

- We propose an SNR Scheduler that dynamically modulates noise levels during training, enabling the model to adaptively learn robust reconstructions across varying corruption intensities in noisy, heterogeneous time series.

- We introduce an SNR Implicit Conditioning Mechanism that adaptively estimates the optimal noise level and denoising steps per input, guiding the reverse process from partially corrupted inputs to achieve faithful and stable reconstructions for accurate TSAD.

- We rectify the limitations of diffusion models in TSAD concerning stochastic generation and substitute three associated, yet separately configured hyperparameters with parameters associated with SNR, thereby presenting a more intuitive and efficient framework for diffusion-based TSAD.

## 2 RELATED WORK

TSAD is pivotal in a variety of domains such as finance, IoT, and cybersecurity. Classical approaches rely on statistical (Li et al., 2022), decomposition (Schölkopf et al., 2001; Liu et al., 2008), or distance-based techniques (He et al., 2003) to identify deviations but struggle with temporally heterogeneous data. Modern deep learning approaches to temporal modeling largely fall into two categories. (i) Representation-based methods learn a manifold of normal data and detect anomalies through measures such as statistical deviations (Ruff et al., 2018; Yang et al., 2023). These methods assume normal data lies in a compact latent region, an assumption frequently violated in practice due to complex, non-stationary time series, resulting in performance degradation. (ii) Reconstruction-based methods learn to reconstruct normal training samples, treating reconstruction error as an anomaly score (Wu et al., 2022; Fan et al., 2023; Xu et al., 2022; Wang et al., 2023; Du et al., 2021; Dai et al., 2024). However, they frequently suffer from training instability (e.g., in GANs) or error propagation (e.g., in autoregressive Transformers). Diffusion-based methods have recently emerged as a promising solution to these issues.

DiffusionAE Pintilie et al. (2023) performs diffusion-based reconstruction on top of the autoencoder's outputs. DiffADT Zuo et al. (2024) employs a state-space model as the denoising network to capture long-term dependencies. MODEM Zhong et al. (2025) incorporates multi-resolution contextual information into a coarse-to-fine diffusion process to refine diffusion trajectories. Imputation-based Xiao et al. (2023); Chen et al. (2023); Cao et al. (2024) frameworks adopt a mask-then-impute paradigm, treating part of the input as an observable conditioning sequence and using a diffusion model to fill in the masked segments for reconstruction. However, these approaches suffer from a critical limitation: they fail to achieve input-consistent reconstruction.

In this work, we adaptively control noise injection via an SNR-aware schedule to preserve critical information, while an SIC mechanism determines optimal noise levels and denoising steps per input, guiding input-consistent reconstruction of sequences. This jointly mitigates the effects of stochasticity on anomaly detection.

## 3 PRELIMINARIES

### 3.1 DENOISING DIFFUSION PROBABILISTIC MODELS (DDPM)

DDPM (Ho et al., 2020) comprises a forward phase that incrementally injects Gaussian noise into input data and a reverse phase that reconstructs samples by iteratively denoising the perturbed representations. Specifically, let $\mathbf{x}_0 \sim q(\mathbf{x}_0)$, the data distribution of interest. The forward process gradually perturbs $\mathbf{x}_0$ over $T$ timesteps by injecting noise scaled by a schedule $\{\beta_t\}_{t-1}^T$, where $0 < \beta_t < 1$, defining a Markov chain:

$$q(\mathbf{x}_{1:T}|\mathbf{x}_0) = \prod_{t=1}^T q(\mathbf{x}_t|\mathbf{x}_{t-1}), \quad \text{with } q(\mathbf{x}_t|\mathbf{x}_{t-1}) = \mathcal{N}\left(\mathbf{x}_t; \sqrt{1-\beta_t}\mathbf{x}_{t-1}, \beta_t \mathbf{I}\right). \quad (1)$$

By recursively applying the reparameterization trick, the state at step $t$ can be expressed in closed form as

$$\mathbf{x}_t = \sqrt{\bar{\alpha}_t}\mathbf{x}_0 + \sqrt{1-\bar{\alpha}_t}\epsilon, \quad \epsilon_t \sim \mathcal{N}(0, \mathbf{I}), \quad (2)$$

where $\bar{\alpha}_t = \Pi_{i=1}^t (1-\beta_i)$. When $t \to T$, $\bar{\alpha}_t \to 0$, and $\mathbf{x}_T$ converges to isotropic Gaussian noise.

The reverse process learns to invert this corruption by training a neural network $\mu_\theta(\mathbf{x}_t, t)$ to approximate the posterior $q(\mathbf{x}_{t-1}|\mathbf{x}_t)$. Starting from $\mathbf{x}_T \sim \mathcal{N}(0, \mathbf{I})$, the model reconstructs $\mathbf{x}_0$ via the

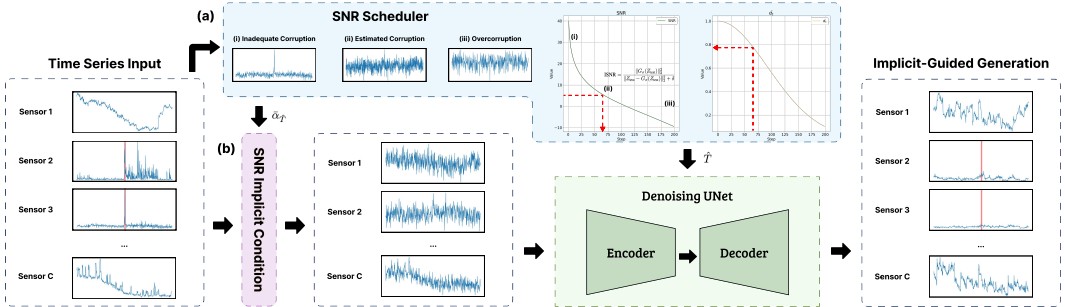

Figure 2: Diagram of ICDiffAD. During inference, the input instance is processed by **(b) the SIC Module** (§4.3), which dynamically estimates (1) the corruption factor $\bar{\alpha}_{\hat{T}}$ and (2) the denoising timestep $\hat{T}$, guided by **(a) the SNR Scheduler** (§4.4). These parameters drive the denoising of implicitly corrupted inputs, enabling input-consistent Implicit-Guided Generation to reconstruct the target signal. As illustrated in the SNR Scheduler visualization, anomalies in one channel are entirely suppressed near an $\text{ISNR}_{\text{dB}}$ value of 5 dB, estimated from Eq.(12).

learned transitions:

$$p_\theta(\mathbf{x}_{t-1}|\mathbf{x}_t) = \mathcal{N}\left(\mathbf{x}_{t-1}; \mu_\theta(\mathbf{x}_t, t), \sigma_t^2 \mathbf{I}\right), \tag{3}$$

where $\sigma_t^2$ is often fixed to $\beta_t$ or $\frac{1-\bar{\alpha}_{t-1}}{1-\bar{\alpha}_t}\beta_t$ for analytical tractability. The joint distribution is thus $p_\theta(\mathbf{x}_{0:T}) = p(\mathbf{x}_T)\Pi_{t=1}^T p_\theta(\mathbf{x}_{t-1}|\mathbf{x}_t)$. The model is trained by minimizing the variational lower bound (VLB) of the negative log-likelihood, which simplifies to a weighted sum of $\mathbb{E}_{\epsilon,t}[||\epsilon - \epsilon_\theta(\mathbf{x}_t, t)||^2]$ when predicting noise residuals $\epsilon$.

## 3.2 DIFFUSION MODELS FOR TSAD

Most existing diffusion-based time series anomaly detection methods employ conditional diffusion models. These approaches partition the data into observed $\mathbf{x}^{\text{con}}$ and masked sequences $\mathbf{x}^{\text{tar}}$, introducing auxiliary information $\mathbf{x}^{\text{con}}$ during training to guide the model in learning to reconstruct the masked sequences. Formally, the goal of these methods is to estimate the true conditional data distribution $q(\mathbf{x}_0^{\text{tar}}|\mathbf{x}_0^{\text{con}})$ using the model distribution $p_\theta(\mathbf{x}_0^{\text{tar}}|\mathbf{x}_0^{\text{con}})$, where the subscript of x denotes the diffusion timestamp with 0 indicating the original data. Let us consider the following distribution (Cao et al., 2024; Tashiro et al., 2021):

$$p_\theta\left(\mathbf{x}_{0:T}^{\text{tar}} \mid \mathbf{x}_0^{\text{con}}\right) = p\left(\mathbf{x}_T^{\text{tar}}\right)\prod_{t=1}^T p_\theta\left(\mathbf{x}_{t-1}^{\text{tar}} \mid \mathbf{x}_t^{\text{tar}}, \mathbf{x}_0^{\text{con}}\right), \quad \mathbf{x}_T^{\text{tar}} \sim \mathcal{N}(0, \mathbf{I}). \tag{4}$$

Correspondingly, the denoising process at step $k$ is given by

$$p_\theta\left(\mathbf{x}_{k-1}^{\text{tar}} \mid \mathbf{x}_k^{\text{tar}}, \mathbf{x}_0^{\text{con}}\right) = \mathcal{N}\left(\mathbf{x}_{k-1}^{\text{tar}}; \mu_\theta\left(\mathbf{x}_k^{\text{tar}}, k \mid \mathbf{x}_0^{\text{con}}\right), \sigma_\theta^2\left(\mathbf{x}_k^{\text{tar}}, k \mid \mathbf{x}_0^{\text{con}}\right)\mathbf{I}\right). \tag{5}$$

Although this approach introduces partial original information and provides some guidance for input-consistent reconstruction, it increases the risk of incorporating anomalous segments as conditional inputs, which may lead to continuous accumulation of erroneous information during the reconstruction process.

## 4 METHODOLOGY

The input to TSAD can be represented as a matrix $X \in \mathbb{R}^{L \times K}$, where $L$ denotes the length of the timestamps and $K$ is the number of variables. The objective is to produce an output vector $\mathbf{y} \in \mathbb{R}^L$, where $y_i \in \{0, 1\}$ indicates whether the observation at the $i$-th timestamp is anomalous.

### 4.1 OVERVIEW

The ICDiffAD is schematized in Figure 2. It incorporates two core innovations that jointly enable input-consistent reconstruction, a property essential for reconstruction-based TSAD methods.

First, we propose to reparameterize the noise scheduler with SNR. For diffusion processes on time series, SNR provides a measurable and intuitive metric of signal energy retention. Our **SNR scheduler** redefines the diffusion noise curriculum by strategically modulating the noise injection rate as a function of SNR, ensuring consistent signal energy retention across datasets with different characteristics. This SNR-calibrated adaptive noise scheduling enhances the model's reconstruction capability under varying degrees of signal energy retention.

Second, our **SIC Mechanism** estimates two critical parameters: the corruption factor $\alpha_{\hat{T}}$, which quantifies the optimal noise intensity for a given input; and the denoising step $\hat{T} \leq T$, which identifies the optimal reverse denoising trajectory. The implicit conditioning controlled by these two key parameters enables the diffusion model to achieve input-consistent generation. Formally, suppose $Z_t$ is the controlled corrupted input at time $t$, the SNR-guided generation is defined as $\hat{Z}_0 = p_\theta(Z_{\hat{T}}) \prod_{t=0}^{\hat{T}} p_\theta(Z_{t-1}|Z_t)$.

The final anomaly detection computes the $L_2$ norm, $s = \|Z_0 - \hat{Z}_0\|_2^2$, between the original time series $Z_0$ and its implicitly guided reconstruction $\hat{Z}_0$ generated through the learned diffusion process. This deviation metric $s$ serves as the anomaly score, with larger values indicating a higher likelihood of temporal anomalies. The formal definition of $Z_0$ will be elaborated in §4.2.

## 4.2 DATA PREPROCESSING

To capture inter-variate dependencies while preserving temporal coherence, we adopt the dimensional transformation from DiffAD (Xiao et al., 2023), reformulating the multivariate time series $X \in \mathbb{R}^{L \times K}$ as a 2D matrix $Z \in \mathbb{R}^{L \times \hat{K}}$ through channel-wise concatenation:

$$Z = \bigoplus_{k=1}^{\hat{K}} X^{(k)} \in \mathbb{R}^{L \times \hat{K}}, \tag{6}$$

where $\bigoplus$ denotes concatenation along the feature dimension. This transformation enables the diffusion model to learn cross-variate correlations through convolutional filters. The detailed algorithm of this transformation is shown in Appendix 8.2.

## 4.3 SNR SCHEDULER

Traditional noise scheduler is parameterized by $\beta_{\min}$, $\beta_{\max}$, and $T$, which independently defines different but related physical dynamics. For example, larger $T$ and smaller $\beta_{\max} - \beta_{\min}$ result in a finer denoising process. However, a larger $T$ might result in a longer inference time, and a smaller $\beta_{\max} - \beta_{\min}$ might cause the corruption process to be insufficient for training a diffusion model.

Hence, we reparameterize these two parameters with SNR, an intuitive, simple, and relatable metric for time series data. Let $\text{TSNR}_{\text{dB}}$ denote the target signal-to-noise ratio as

$$\text{TSNR}_{\text{dB}} = 10 \log_{10} \frac{M_T}{M_0 - M_T}, \tag{7}$$

where $M_T$ is that the terminal signal energy and $M_0 = \mathbb{E}[\|Z_0\|_F^2]$. Doing so will ensure that the final SNR of the noise scheduler exactly matches the user-specified $\text{TSNR}_{\text{dB}}$. We then define $M_t = \alpha_t M_{t-1}$ with $\alpha_t \in (0, 1)$. The recursive relation gives

$$M_T = M_0 \prod_{t=1}^{T} \alpha_t \implies \log M_T - \log M_0 = \sum_{t=1}^{T} \log \alpha_t. \tag{8}$$

This recursive relation ensures that the total energy dissipated across all diffusion steps equals the predefined TSNR, guaranteeing that terminal noise levels match the target. To ensure monotonic energy decay ($M_t < M_{t-1}, \forall t$), we introduce a temporal corruption scheduler $g : \mathbb{Z}^+ \to \mathbb{R}^+$ (e.g., linear scheduler $g(t) = t$) that explicitly modulates phase-space trajectory dispersion:

$$\alpha_t = \exp\left(\frac{\log(M_T/M_0)}{\sum_{s=1}^{T} g(s)} \cdot g(t)\right). \tag{9}$$

This parameterization of $\{\alpha_t\}$ through $\text{TSNR}_{\text{dB}}$ achieves two critical advancements over conventional approaches. First, we eliminate heuristic tuning of $\beta_{\min}$, $\beta_{\max}$, and $T$ by deriving them from the physically interpretable $\text{TSNR}_{\text{dB}}$. Second, by introducing a controllable noise scaling, we enable quantifiable noise injection at each diffusion step via the instantaneous SNR metric ($\text{SNR}(t) \leq \text{TSNR}_{\text{dB}}, t \in [0, T]$):

$$\text{SNR}(t) = \frac{\bar{\alpha}_t}{1 - \bar{\alpha}_t} < 1, \quad \bar{\alpha}_t = \prod_{s=1}^{t} \alpha_s, \tag{10}$$

where $\bar{\alpha}_t$ represents the cumulative signal retention up to step $t$.

## 4.4 SNR IMPLICIT CONDITIONING

The inherent stochasticity of diffusion models poses significant challenges to TSAD, where input-consistent reconstruction of normative patterns is a prerequisite for accurate residual-based anomaly scoring. Conventional denoising processes permit multiple trajectory realizations $\{Z_t^{(i)}\}_{i=1}^N$ from a single noise instance $Z_T$, leading to irreducible variance in anomaly scores $s(Z_0) = \mathbb{E}_\theta[\|Z_0 - \hat{Z}_0\|_2^2]$. To mitigate this, we introduce SIC, a spectral alignment mechanism that estimates the optimal corruption level for each input instance during inference.

SIC mechanism identifies the optimal corruption factor and denoising step by estimating the amount of original signal information that is sufficient to guide the diffusion model toward input-consistent generation. The SNR provides a physically interpretable measure of signal fidelity across frequency bands. By modulating the SNR, we obtain an implicit conditioning that guides the diffusion model to input-consistent reconstruction while preserving its flexibility. Let $Z_{\text{test}} \in \mathbb{R}^{W \times K}$ denote a test instance with $W$ timesteps and $K$ variates. We first decompose $Z_{\text{test}}$ via spectral Gaussian filtering:

$$Z_{\text{test}} = \underbrace{Z_{\text{con}}}_{\text{Low Frequency}} + \underbrace{N}_{\text{Residual}}, \quad Z_{\text{con}} = G_\sigma(Z_{\text{test}}), \quad N = Z_{\text{test}} - G_\sigma(Z_{\text{test}}), \tag{11}$$

where $G_\sigma : \mathbb{R}^W \to \mathbb{R}^W$ is a zero-phase Gaussian low-pass filter with bandwidth $\sigma$. The residual $N$ contains all components other than the low-frequency part. From this decomposition, we estimate the inference signal-to-noise ratio (ISNR), an instance-specific metric of signal energy retention that quantifies the optimal noise corruption for implicit conditioning:

$$\text{ISNR} = \frac{\|Z_{\text{con}}\|_F^2}{\|N\|_F^2 + \delta} = \frac{\|G_\sigma(Z_{\text{test}})\|_F^2}{\|Z_{\text{test}} - G_\sigma(Z_{\text{test}})\|_F^2 + \delta}, \tag{12}$$

where $\delta > 0$ prevents division by zero in noiseless edge cases. The Frobenius norm aggregates energy across all $K$ variates. The ISNR guides the adaptive initialization of the reverse diffusion process through two estimation phases: estimating the optimal noise intensity $\alpha^*$ from ISNR; and aligning $\alpha^*$ to the nearest diffusion step.

**Phase 1: Optimal Corruption Factor Estimation** Given the instantaneous ISNR from Eq.(12), we derive the instance-optimal corruption factor $\alpha^*$ leveraging the statistical properties of $Z_{\text{test}}$ (see detailed proof in Appendix 8.1):

$$\alpha^* = \frac{\text{ISNR}}{\text{ISNR} + \mu^2(Z_{\text{test}}) + \sigma^2(Z_{\text{test}})}, \tag{13}$$

where $\mu^2(Z_{\text{test}})$ and $\sigma^2(Z_{\text{test}})$ denote the mean and variance computed for elements in the test instance $Z_{\text{test}}$, respectively.

**Phase 2: Diffusion Step Alignment** Since $\alpha^*$ may not reside in the pretrained schedule $\{\bar{\alpha}_t\}_{t=1}^T$, we project it onto the nearest feasible step $\hat{T}$:

$$\hat{T} = \arg\min_{t \leq T} |\bar{\alpha}_t - \alpha^*|. \tag{14}$$

Finally, the test instance $Z_{\text{test}}$ is then corrupted to $Z_{\text{test}}^{\text{con}} = Z_{\hat{T}} = \sqrt{\bar{\alpha}_{\hat{T}}} Z_{\text{test}} + \sqrt{1 - \bar{\alpha}_{\hat{T}}} \epsilon$, where Gaussian noise $\epsilon$ strategically degrades anomalous components while preserving original data correlations. This controlled initialization enables a conditional reverse process governed by Eq.(15) while

Table 1: Results on the five real-world datasets. For each metric, the higher the value, the better the performance. All values are the average percentages over five random seeds. The best F1 score is highlighted in **bold** and the second best is underlined.

| Dataset | Method | MSL P | MSL R | MSL F1 | SMAP P | SMAP R | SMAP F1 | SMD P | SMD R | SMD F1 | PSM P | PSM R | PSM F1 | SWaT P | SWaT R | SWaT F1 | Average F1 |
|---|---|---|---|---|---|---|---|---|---|---|---|---|---|---|---|---|---|
| Classical | IF | 14.87 | 78.10 | 24.98 | 13.46 | 92.37 | 23.50 | 14.30 | 23.94 | 17.91 | 32.82 | 89.67 | 48.05 | 99.74 | 58.61 | 73.83 | 37.65 |
| | ECOD | 16.55 | 45.61 | 24.29 | 13.22 | 97.80 | 23.29 | 13.62 | 19.93 | 16.18 | 32.79 | 84.89 | 47.30 | 97.70 | 59.63 | 74.06 | 37.03 |
| | CBLOF | 15.21 | 68.00 | 24.86 | 17.93 | 59.44 | 27.55 | 16.89 | 23.58 | 19.68 | 33.06 | 94.81 | 49.03 | 99.46 | 58.61 | 73.75 | 38.98 |
| | PCA | 11.29 | 64.04 | 19.20 | 13.19 | 89.75 | 23.00 | 15.63 | 20.59 | 17.77 | 31.13 | 92.78 | 46.62 | 99.32 | 62.30 | 76.57 | 36.63 |
| Representation | Deep SVDD | 12.94 | 16.04 | 14.32 | 11.77 | 17.09 | 13.94 | 14.99 | 15.78 | 15.37 | 36.63 | 36.30 | 36.46 | 81.29 | 61.95 | 70.31 | 30.08 |
| | DCDetector | 2.35 | 3.46 | 2.80 | 8.04 | 11.30 | 9.40 | 3.71 | 9.03 | 5.26 | 28.97 | 12.05 | 17.02 | 46.45 | 36.36 | 40.79 | 15.05 |
| Recontruction | ATF-UAD | 15.42 | 34.60 | 21.33 | 12.79 | 99.99 | 22.68 | 6.07 | 24.06 | 9.49 | 30.50 | 91.67 | 45.77 | 30.09 | 73.37 | 42.68 | 28.39 |
| | AT | 10.54 | 100.00 | 19.07 | 12.79 | 100.00 | 22.68 | 4.15 | 100.00 | 7.98 | 27.73 | 100.00 | 43.42 | 12.02 | 100.00 | 21.46 | 22.92 |
| | FGANomaly | 10.55 | 99.56 | 19.08 | 12.79 | 100.00 | 22.68 | 13.65 | 19.90 | 16.19 | 30.58 | 92.76 | 46.00 | 83.99 | 59.77 | 69.84 | 34.76 |
| | TimesNet | 13.44 | 84.26 | 23.18 | 14.03 | 88.41 | 24.22 | 15.42 | 33.41 | 21.10 | 27.74 | 100.00 | 43.43 | 12.14 | 100.00 | 21.65 | 26.72 |
| | D3R | 11.59 | 67.18 | 19.77 | 13.36 | 82.10 | 22.97 | 15.51 | 29.66 | 20.37 | 28.01 | 97.33 | 43.50 | 90.56 | 23.74 | 26.07 |
| | SARAD | 10.74 | 91.94 | 19.24 | 12.79 | 100.00 | 22.68 | 17.06 | 41.26 | **24.10** | 41.55 | 58.60 | 48.60 | 96.20 | 66.92 | **78.92** | 38.71 |
| Diffusion Model | DiffAD | 10.53 | 100.00 | 19.06 | 12.80 | 99.67 | 22.68 | 10.60 | 22.08 | 14.32 | 27.87 | 98.88 | 43.48 | 12.14 | 100.00 | 21.65 | 24.24 |
| | ImDiffusion | 34.43 | 9.72 | 15.16 | 8.60 | 1.58 | 2.67 | 11.95 | 11.54 | 11.74 | 66.62 | 5.07 | 9.42 | NA | NA | NA | 9.75 |
| | **Ours** | 26.16 | 45.33 | **33.17** | 16.47 | 93.61 | **28.01** | 22.64 | 24.82 | 23.68 | 45.32 | 81.34 | **56.87** | 99.38 | 63.30 | 77.34 | **43.81** |

ensuring training-inference SNR consistency and anomaly attenuation:

$$p_\theta \left( Z_{0:\hat{T}} \mid Z_{\text{test}}^{\text{con}} \right) = p_\theta(Z_{\hat{T}}) \prod_{k=1}^{\hat{T}} p_\theta \left( Z_{k-1} \mid Z_k \right). \tag{15}$$

## 5 EXPERIMENTS

### 5.1 EXPERIMENTAL SETUP

**Datasets**  We evaluate our method on five real-world datasets collected from industrial control and service monitoring scenarios: Mars Science Laboratory rover (MSL) (Hundman et al., 2018), Soil Moisture Active Passive satellite (SMAP) (Hundman et al., 2018), Server Machine Dataset (SMD) (Su et al., 2019), Pooled Server Metrics (PSM) (Abdulaal et al., 2021), and Secure Water Treatment (SWaT) (Mathur and Tippenhauer, 2016). Details of these datasets are provided in Appendix 8.3.1.

**Evaluation Metrics**  For performance evaluation, we report standard Precision (P), Recall (R), and F1 scores. To avoid potential overestimation introduced by point-adjustment methods (Sarfraz et al., 2024; Kim et al., 2022), we adopt a strict evaluation protocol based on exact match of predicted and ground-truth anomaly points.

**Baselines**  We conduct extensive comparisons between our model and a range of state-of-the-art anomaly detection baselines. Classical machine learning methods such as Isolation Forest (IF) (Liu et al., 2008), ECOD (Li et al., 2022), CBLOF (He et al., 2003) and PCA (Schölkopf et al., 2001) are included to represent traditional approaches. We also evaluate reconstruction-based deep-learning methods, including the CNN-based TimesNet (Wu et al., 2022), the Transformer-based models such as Anomaly Transformer (AT) (Xu et al., 2022), ATF-UAD (Fan et al., 2023), D3R (Wang et al., 2023) and Sarad (Dai et al., 2024), the GAN-based FGANomaly (Du et al., 2021), and the diffusion-based methods such as DiffAD (Xiao et al., 2023) and ImDiffusion (Chen et al., 2023). In addition, we consider representation learning methods, including Deep SVDD (Ruff et al., 2018) and DCDetector (Yang et al., 2023). Notably, DiffAD and ImDiffusion leverage the strong generative capabilities of diffusion models to model temporal dependencies for anomaly detection. All baselines are implemented using their officially released code, with hyperparameters set according to the recommended configurations provided in the original papers.

### 5.2 RESULTS

As shown in Table 1, ICDiffAD achieves significant improvements across five benchmark datasets, demonstrating two critical advances. First, **ICDiffAD outperforms previous diffusion-based methods** by substantial margins, with +19.57% F1 improvement over DiffAD and +34.06% over

Table 2: Ablation study for component contributions. The standard F1 scores are reported.

| IC | SNR | SIC | MSL | SMAP | SMD | PSM | SWaT | Avg |
|----|-----|-----|------|------|------|------|------|------|
| ✗ | ✗ | ✗ | 23.01 | 22.81 | 16.16 | 44.57 | 76.05 | 36.52 |
| ✓ | ✗ | ✗ | 31.88 | 23.01 | 21.14 | 48.65 | 76.69 | 40.27 |
| ✓ | ✓ | ✗ | 32.60 | 26.27 | 22.84 | 54.80 | 77.19 | 42.74 |
| ✓ | ✓ | ✓ | **33.17** | **28.01** | **23.68** | **56.87** | **77.34** | **43.81** |

Table 3: Ablation study for $g(t)$. The standard F1 scores are reported.

| $g(t)$ | MSL | SMAP | SMD | PSM | SWaT |
|--------|------|------|------|------|------|
| Linear | 33.17 | 28.01 | 23.68 | 56.87 | 77.34 |
| Quadratic | 34.78 | 27.70 | 27.42 | 55.97 | 77.33 |
| Cosine | 34.81 | 26.76 | 26.16 | 55.54 | 77.33 |

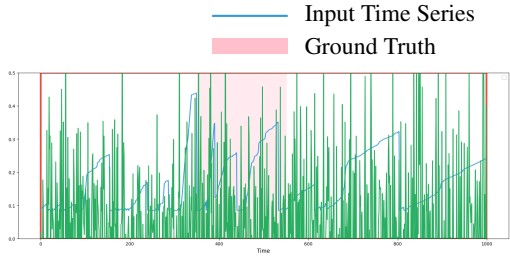

(a) stochastic generation by DiffAD

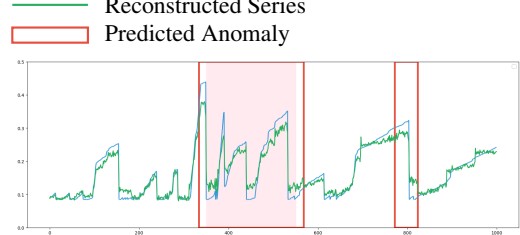

(b) input-consistent generation by ICDiffAD

Figure 3: Visualization of the results obtained by DiffAD (Xiao et al., 2023) and ICDiffAD on MSL. The region of pale-pink shadow indicates the ground truth of anomaly events.

ImDiffusion, validating that our SNR-guided input-consistent generation (via the SIC module) resolves the reconstruction instability of vanilla diffusion models for TSAD. The catastrophic failure of ImDiffusion (-14.49% F1 vs. DiffAD) further underscores the necessity of our SNR-guided implicit conditioning approach. Second, **ICDiffAD surpasses SARAD**, the prior reconstruction-based state-of-the-art, by +5.1% average F1, achieving an 8.27% absolute improvement on the noisy PSM dataset despite minor gaps in SMD (-0.42%) and SWaT (-1.58%). This highlights the SNR Scheduler's effectiveness in handling temporal heterogeneity. As shown in Appendix 8.4.1, ICDiffAD reduces false positives (-60.23%) versus DiffAD while maintaining competitive recall, achieving better precision-recall balance. Full metrics (AUC-ROC, AUC-PR, VUS-ROC, VUS-PR (Paparrizos et al., 2022), Affiliation-based metrics (Huet et al., 2022), point-adjustment metrics), standard deviations of Table 1, and model complexity are detailed in Appendices 8.4.2- 8.4.4. Figure 3 compares the reconstruction and anomaly detection of DiffAD (Figure 3a) and ICDiffAD (Figure 3b). ICDiffAD not only shows a more faithful reconstruction, but it also successfully attenuates the anomalous signals, resulting in better anomaly detection performance.

Despite current advancements in deep learning, classical methods exhibit strong performance in our experiments. In particular, on the SWaT dataset, IForest and PCA significantly outperform deep learning methods. This is because most anomalies in SWaT can be easily detected through naive methods, despite complex temporal heterogeneity. However, experiments with SMD further prove our case when dealing with complex temporal heterogeneity and anomalies, as classical methods are being surpassed by deep learning methods such as TimesNet and SARAD. ICDiffAD introduces two improvements for current diffusion models: the SNR Scheduler, which enhances the learning process on different noise scales, and the SIC module, which enables input-consistent generation for TSAD. The synergy between these two modules enables the improved diffusion for TSAD.

## 5.3 Ablation Studies

**Component Contributions** Table 2 quantifies the incremental benefits of our proposed components: SNR Scheduler and SNR Implicit Conditioning (SIC). The vanilla diffusion model (all components disabled) achieves average F1-score of 36.52%, with particularly poor performance on SMD (16.16%) due to its complex temporal heterogeneity. Introducing Implicit Conditioning (IC) alone improves the average F1-score by 3.75%, showing the need for input-consistent generation of Diffusion Models. Coupling IC with the SNR Scheduler yields an additional 2.47 average point gain, as training becomes more guided by SNR, rather than generation-oriented. Our SIC method estimates the optimal corruption factor and denoising step for diffusion models, achieving peak performance (43.89% average F1) through ISNR-guided anomaly attenuation and input-consistent generation.

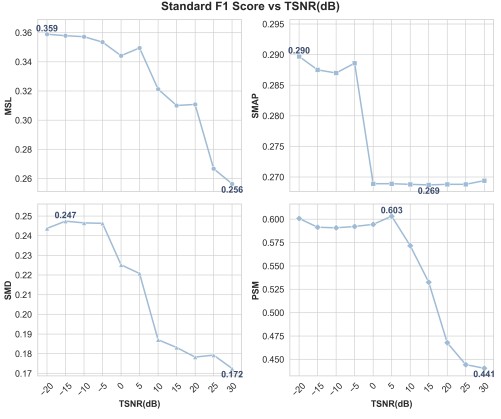

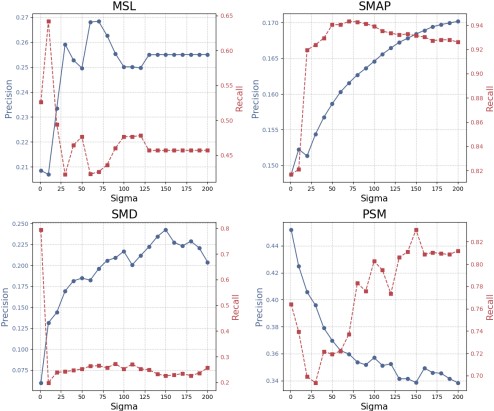

Figure 4: Ablation study of TSNR$_{dB}$ in terms of F1 score

Figure 5: Ablation study of $\sigma$ in terms of precision and recall

**Effect of $g(t)$**    Table 3 compares different choices of $g(t)$ of the SNR Scheduler and demonstrates that the proposed SNR Scheduler is robust to different $g(t)$. This is because the SNR scheduler regulates noise injection according to the specific requirements of the anomaly detection, independent of $g(t)$, which governs the overall trend of noise injection.

**Effect of TSNR$_{dB}$**    The ablation study in Figure 4 empirically demonstrates the operational trade-offs governed by TSNR selection under fixed diffusion steps ($T = 200$). High TSNR levels (15-30 dB) correspond to minimal noise injection during training, inadequately conditioning the model for TSAD and yielding suboptimal F1 scores across all datasets. Conversely, aggressive noise corruption at low TSNR ($\leq 20$ dB) guarantees that the model learns at different noise scale. This mechanism drives consistent F1 improvements as the TSNR decreases, with MSL, SMD, and PSM achieving gains of +10.3, +7.5, and +16.2, respectively, between 30 dB and -20 dB.

The dataset-specific response further empirically substantiates the SNR Scheduler's necessity. MSL and SMD exhibit gradual performance improvements (MSL: $0.2564 \rightarrow 0.3589$; SMD: $0.1725 \rightarrow 0.2437$), reflecting their sensitivity to multiscale anomalies (low-amplitude drifts and high-frequency spikes). In contrast, SMAP and SWaT display abrupt saturation. This is because their time series anomalies have much simpler patterns. Beyond TSNR $<$-20 dB, performance plateaus as the scheduler's adaptive noise curriculum inherently balances coarse and fine-grained generation.

**Effect of $\sigma$**    The Gaussian filter bandwidth $\sigma$ governs the spectral decomposition fidelity of the SIC module, directly influencing the trade-off between normative pattern preservation and anomalous deviation suppression. Figure 5 quantifies precision ($P$) and recall ($R$) across four TSAD benchmarks spanning heterogeneous temporal entropy ($H$): PSM ($H = 6.44$), MSL ($H = 4.06$), SMD ($H = 3.83$), and SMAP ($H = 3.05$).

**For Complex Temporal Regimes** (e.g., PSM), ICDiffAD achieves maximal performance ($P = 0.45$, $R = 0.76$) at minimal $\sigma = 1$. At $\sigma = 1$, aggressive high-pass filtering isolates high-frequency into the residual $N$, minimizing ISNR. This induces a small corruption factor, enforcing the reverse diffusion process to start from a lightly corrupted signal and retaining most of the complex temporal information. Consequently, input-consistent reconstruction dominates. Further increasing $\sigma$ will degrade the performance on PSM, because stochastic reconstruction is suboptimal in complex temporal regimes.

**For Simpler Temporal Patterns** (MSL, SMAP, and SMD), increasing $\sigma$ enhances precision (MSL: $P = 0.21 \rightarrow 0.25$; SMAP: $P = 0.21 \rightarrow 0.67$; SMD: $P = 0.18 \rightarrow 0.52$) but reduces recall (MSL: $R = 0.53 \rightarrow 0.46$; SMAP: $R = 0.89 \rightarrow 0.34$; SMD: $R = 0.91 \rightarrow 0.41$). Larger $\sigma$ elevates ISNR, increasing the denoising factor and resulting in the retention of the coarse trends. While this suppresses high-frequency noise (improving precision), it attenuates reconstruction errors for subtle anomalies embedded in low-frequency drifts (reducing recall).

## 6 CONCLUSION

We propose ICDiffAD, a diffusion-based time series anomaly detection method that innovatively introduces an SNR-guided noise scheduler and an implicit conditioning mechanism to address the challenge of stochastic reconstruction in diffusion models for TSAD. This approach achieves a balance between generative flexibility and reconstruction fidelity. ICDiffAD achieves improvements by +19.57% average F1 score over previous diffusion-based methods and +5.1% average F1 score over reconstruction-based methods, demonstrating its ability to learn robust temporal patterns and detect anomalies through input-consistent generation.

## 7 ACKNOWLEDGEMENTS

This work was supported in part by the National Key R&D Program of China (No.2023YFB4302200) and the Special Foundations for the Development of Strategic Emerging Industries of Shenzhen(No.KJZD20231023094700001).

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

## 8 APPENDIX

### 8.1 PROOF OF CORRUPTION FACTOR ESTIMATION

In this section, we provide the proof of Eq.(13) presented in this paper.

According to Eq.(2), the noise corruption $Z_t$ after $t$ steps in the diffusion process is given by

$$Z_t = \sqrt{\bar{\alpha}_t} Z_0 + \sqrt{1 - \bar{\alpha}_t}\, \epsilon_t \tag{16}$$

For $Z_t$, the term $\sqrt{\bar{\alpha}_t} Z_0$ represents the useful signal, while $\sqrt{1 - \bar{\alpha}_t}\, \epsilon_t$ corresponds to the noise component. The signal-to-noise ratio $\text{SNR}_t$ is therefore given by

$$\text{SNR}_t = \frac{\mathbb{P}\left(\sqrt{\bar{\alpha}_t}\, Z_0\right)}{\mathbb{P}\left(\sqrt{1 - \bar{\alpha}_t}\, \varepsilon_t\right)}. \tag{17}$$

where $\mathbb{P}$ denotes the power operator, which is specifically defined as

$$\mathbb{P}(Z) = \mu^2(Z) + \sigma^2(Z), \tag{18}$$

where $\mu(Z)$ and $\sigma^2(Z)$ denote the mean and variance computed for elements in the $Z$, respectively. Thus,

$$\text{SNR}_t = \frac{\mu^2\left(\sqrt{\bar{\alpha}_t}\, Z_0\right) + \sigma^2\left(\sqrt{\bar{\alpha}_t}\, Z_0\right)}{\mu^2\left(\sqrt{1 - \bar{\alpha}_t}\, \epsilon_t\right) + \sigma^2\left(\sqrt{1 - \bar{\alpha}_t}\, \epsilon_t\right)}. \tag{19}$$

As $\epsilon \sim \mathcal{N}(0, \mathbf{I})$, it follows that

$$\text{SNR}_t = \frac{\bar{\alpha}_t \left(\mu^2(Z_0) + \sigma^2(Z_0)\right)}{1 - \bar{\alpha}_t}. \tag{20}$$

Therefore, we estimate the corruption factor in the denoising process based on the ISNR as

$$\alpha^* = \frac{\text{ISNR}}{\text{ISNR} + \mu^2(Z_0) + \sigma^2(Z_0)}. \tag{21}$$

### 8.2 DATA PROCESSING PSEUDOCODE

```python
# Step 1: Load original training data
X_input = np.load("path/to/data")

# Step 2: Normalize training data (Standard Process)
X_norm = normalize(X_input)   # minmax scaler on every channel

# Step 3: Extend features (repeat first C channels)
X_extra = X_norm[:, :C]
X_extended = concatenate(X_norm, X_extra, axis=1)

# Step 4: Slice into sliding windows
for i in range(0, len(X_extended) - win_size + 1, win_size):
    window = X_extended[i : i + win_size]

    # Step 5: Format window
    Z = unsqueeze(window)  # shape: (1, win_size, num_features)

    # Step 6: Output
    yield Z
```

### 8.3 DETAILED EXPERIMENTAL SETTINGS

#### 8.3.1 DATASETS

We evaluate anomaly detection performance using five real-world datasets spanning diverse application domains, including industrial control and IT service monitoring. The anomalies in these datasets range from IT service outages to external cyber-physical attacks targeting control systems.

Table 4: Statistics of the main datasets

| Datasets | Domain | Entities | Dimension | Train # | Test # | Anomaly Ratio |
|----------|--------|----------|-----------|---------|--------|---------------|
| MSL | Spacecraft | 27 | 55 | 58,317 | 73,729 | 10.50% |
| SMAP | Spacecraft | 55 | 25 | 135,183 | 427,617 | 12.80% |
| SMD | Server Machine | 38 | 38 | 708,405 | 708,420 | 4.20% |
| PSM | Server Machine | 25 | 25 | 132,481 | 87,841 | 27.80% |
| SWaT | Water Treatment | 51 | 51 | 496,800 | 449,919 | 12.10% |

- **Server Machine Dataset(SMD)** (Su et al., 2019): The SMD dataset was collected from a large-scale internet company and contains five weeks of data from 28 server machines, each equipped with 38 sensors. The first five days consist solely of normal data, while the last five days intermittently include anomalies.

- **Pooled Server Metrics(PSM)** (Abdulaal et al., 2021): The PSM dataset was collected from internal nodes of multiple application servers at eBay. It contains 13 weeks of training data and 8 weeks of test data. Anomalies are present in both sets, but labels are provided only for the test set. These labels were manually created by engineers and domain experts, and may include both scheduled and unscheduled anomalies.

- **Mars Science Laboratory(MSL) and Soil Moisture Active Passive(SMAP)** (Hundman et al., 2018): The MSL and SMAP datasets are publicly available datasets collected by NASA. They consist of telemetry anomaly data derived from Incident Surprise Anomaly (ISA) reports in spacecraft monitoring systems. The MSL dataset contains 55 dimensions, while SMAP has 25 dimensions. The training sets for both datasets contain unlabeled anomalies.

- **Secure Water Treatment(SWaT)** (Mathur and Tippenhauer, 2016):The SWaT dataset was collected from a scaled-down water treatment testbed instrumented with 51 sensors over 11 days. The first 7 days contain only normal data. In the last 4 days, 41 anomalies were injected using various attack strategies.

Table 4 summarizes the statistics of these datasets. Train# and Test# denote the number of training and testing samples, respectively. The anomaly rate is defined as the ratio of the number of anomalous points to the total number of test samples.

### 8.3.2 BASELINE

All baselines are evaluated based on our own runs using the same hardware environment. We trained all baseline models using their official or open-source implementations and followed the hyperparameter settings recommended in their respective papers. The publicly accessible URLs of the baselines used are listed as follows:

- IF (Liu et al., 2008): https://github.com/yzhao062/pyod
- ECOD (Li et al., 2022): https://github.com/yzhao062/pyod
- CBLOF (He et al., 2003): https://github.com/yzhao062/pyod
- PCA (Schölkopf et al., 2001): https://github.com/yzhao062/pyod
- DeepSVDD (Ruff et al., 2018): https://github.com/yzhao062/pyod
- DCDetector(Yang et al., 2023): https://github.com/DAMO-DI-ML/KDD2023-DCdetector
- ATF-UAD (Fan et al., 2023): https://github.com/wzhSteve/ATF-UAD
- AT (Xu et al., 2022): https://github.com/thuml/Anomaly-Transformer
- FGANomaly (Du et al., 2021): https://github.com/sxxmason/FGANomaly
- TimesNet (Wu et al., 2022): https://github.com/thuml/Time-Series-Library
- D3R (Wang et al., 2023): https://github.com/ForestsKing/D3R
- SARAD (Dai et al., 2024): https://github.com/daidahao/SARAD
- DiffAD (Xiao et al., 2023): https://github.com/ChunjingXiao/DiffAD
- ImDiffusion (Chen et al., 2023): https://github.com/17000cyh/IMDiffusion

### 8.3.3 IMPLEMENTATION DETAILS

We implemented ICDiffAD in Python using the PyTorch library and the Hydra framework. All experiments were conducted on a single NVIDIA 3090 GPU (24 GB). We used the Adam optimizer with an initial learning rate of 1e-4. In our experiments, for all datasets, the sliding window size was fixed at 1024, the number of diffusion timesteps was set to 200, the SNR scheduler parameter $TSNR_{\text{dB}}$ was set to -10, and the function $g(t)$ was defined as a linear function. For corruption factor estimation in the SIC mechanism, we performed TPE sampling over Gaussian kernel sizes $s \in \{3, 1024\}$ and Gaussian kernel standard deviations $\sigma \in \{1, 100\}$ on each dataset, and we reported the results corresponding to the highest F1 score.

## 8.4 ADDITIONAL EXPERIMENT RESULTS

### 8.4.1 CONFUSION MATRIX COMPARING DIFFAD AND ICDIFFAD

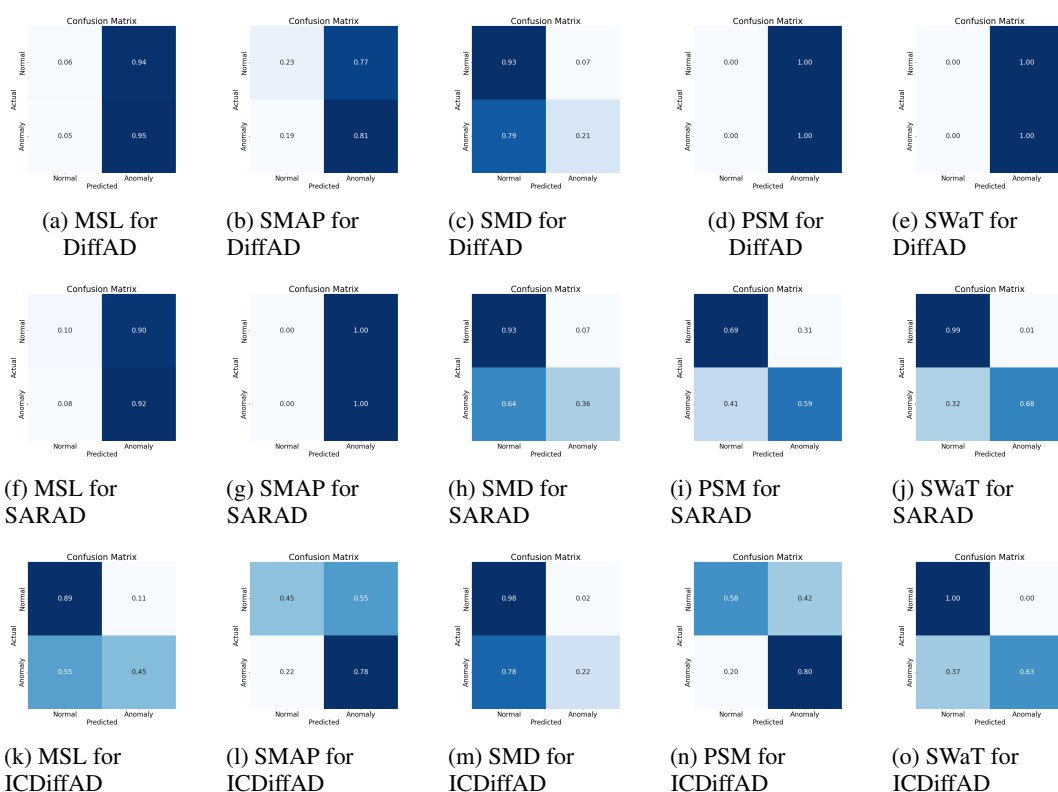

Figure 6: Confusion Matrix Comparing DiffAD, SARAD, and ICDiffAD

Figure 6 shows the confusion matrices of DiffAD (Diffusion Models), SARAD (Reconstruction Models), and ICDiffAD (Ours) for different datasets. Compared with DiffAD, ICDiffAD's SIC mechanism has reduced the average False Positive (Predicted Anomaly but Normal) by 60.2% while maintaining high detection accuracy. The reconstruction-based method struggles in noisy datasets such as PSM, causing the model to have 2 times more False Negatives (Predicted Normal but Anomaly) than ICDiffAD. On the MSL and SMAP datasets, ICDiffAD also demonstrates excellent detection performance with low false positive rates, highlighting its strong robustness in satellite telemetry scenarios. However, our method still faces challenges when dealing with datasets like SMD, which exhibit high temporal heterogeneity. The presence of rapid distributional shifts and diverse temporal patterns in SMD makes it difficult for the model to maintain stable performance across all time periods. Addressing these challenges remains an important avenue for future research in diffusion-based TSAD. Overall, ICDiffAD significantly reduces false alarms, which on the one hand validates the effectiveness of the SIC mechanism in addressing uncertainty reconstruction in

diffusion-based TSAD, and on the other hand, highlights the great potential of diffusion models waiting to be further explored.

### 8.4.2 ADDITIONAL DETECTION RESULTS

Table 5: AUC-ROC and AUC-PR scores across five real-world datasets, where higher values indicate better performance. All values are average percentages from five random seeds. The best results are shown in **bold**, and the second-best results are underlined.

| Dataset Method | | MSL | | SMAP | | SMD | | PSM | | SWAT | | Average | |
|---|---|---|---|---|---|---|---|---|---|---|---|---|---|---|
| | | AUC-ROC | AUC-PR | AUC-ROC | AUC-PR | AUC-ROC | AUC-PR | AUC-ROC | AUC-PR | AUC-ROC | AUC-PR | AUC-ROC | AUC-PR |
| Classical | IF | 60.39 | 14.10 | **62.56** | **15.51** | 66.13 | 12.59 | 70.49 | 45.61 | 83.61 | 72.78 | 68.64 | 32.12 |
| | ECOD | 54.41 | 14.41 | 39.73 | 10.29 | 62.60 | 10.74 | 64.78 | 39.77 | 86.12 | 75.71 | 61.53 | 30.18 |
| | CBLOF | 61.73 | 18.23 | 57.59 | 15.24 | 66.66 | 12.80 | 70.39 | 51.27 | **87.75** | **76.69** | 68.82 | **34.85** |
| | PCA | 49.67 | 12.95 | 41.30 | 10.54 | 64.70 | 10.78 | 65.28 | 47.04 | 81.82 | 72.60 | 60.55 | 30.78 |
| Representation | Deep SVDD | 53.13 | 14.03 | 41.30 | 10.53 | 63.29 | 10.19 | 66.72 | 47.00 | 81.77 | 72.30 | 61.24 | 30.81 |
| | DCDetector | 50.24 | 10.80 | 50.42 | 13.00 | 49.17 | 4.22 | 49.74 | 24.89 | 51.52 | 14.75 | 50.22 | 13.53 |
| Recontruction | ATF-UAD | 57.89 | 16.81 | 40.80 | 10.16 | 52.23 | 4.71 | 57.70 | 38.35 | 62.47 | 17.89 | 54.22 | 17.58 |
| | AT | 49.04 | 9.90 | 51.30 | 14.19 | 49.97 | 4.55 | 45.79 | 26.21 | 42.23 | 10.23 | 47.67 | 13.02 |
| | FGANomaly | 48.75 | 10.31 | 45.05 | 11.35 | 69.65 | 9.69 | 66.52 | **53.21** | 84.94 | 73.60 | 62.98 | 31.63 |
| | TimesNet | 60.17 | 14.29 | 43.32 | 11.35 | 75.57 | **15.22** | 60.09 | 39.32 | 24.13 | 8.60 | 52.66 | 17.76 |
| | D3R | 55.99 | 13.96 | 46.43 | 12.03 | 71.95 | 14.37 | 57.72 | 37.07 | 52.97 | 12.73 | 57.01 | 18.03 |
| | SARAD | 52.72 | 15.65 | 40.91 | 12.39 | **79.82** | 15.1 | 65.42 | 43.28 | 86.91 | 74.81 | 65.16 | 32.25 |
| Diffusion Model | DiffAD | 52.37 | 13.96 | 52.58 | 13.23 | 61.93 | 8.36 | 55.09 | 27.87 | 20.44 | 11.7 | 48.48 | 15.02 |
| | **Ours** | **75.78** | **26.05** | 55.30 | 13.30 | 74.82 | 14.85 | **71.61** | 41.25 | 82.15 | 72.91 | **71.93** | 33.67 |

Table 6: Standard deviations of anomaly detection performance in Table 5. Standard deviations of AUC-ROC and AUC-PR scores are reported. All values are percentages.

| Dataset Method | | MSL | | SMAP | | SMD | | PSM | | SWAT | |
|---|---|---|---|---|---|---|---|---|---|---|---|
| | | AUC-ROC | AUC-PR | AUC-ROC | AUC-PR | AUC-ROC | AUC-PR | AUC-ROC | AUC-PR | AUC-ROC | AUC-PR |
| Classical | IF | 0.96 | 0.33 | 0.98 | 0.37 | 0.72 | 1.50 | 1.12 | 1.70 | 0.91 | 0.86 |
| | ECOD | 0.00 | 0.00 | 0.00 | 0.00 | 0.00 | 0.00 | 0.00 | 0.00 | 0.00 | 0.00 |
| | CBLOF | 1.94 | 0.80 | 0.28 | 0.14 | 0.99 | 1.04 | 0.40 | 1.34 | 0.21 | 0.18 |
| | PCA | 0.00 | 0.00 | 0.00 | 0.00 | 0.00 | 0.00 | 0.00 | 0.00 | 0.00 | 0.00 |
| Representation | Deep SVDD | 2.05 | 0.69 | 1.08 | 0.25 | 2.04 | 1.04 | 5.01 | 5.95 | 0.81 | 0.58 |
| | DCDetector | 0.19 | 0.14 | 0.87 | 0.43 | 0.78 | 0.13 | 0.32 | 0.15 | 0.04 | 0.21 |
| Recontruction | ATF-UAD | 4.09 | 2.61 | 8.69 | 4.20 | 1.21 | 0.25 | 4.88 | 3.10 | 15.50 | 6.36 |
| | AT | 1.32 | 0.97 | 1.37 | 1.26 | 0.44 | 0.43 | 1.95 | 1.01 | 5.39 | 1.95 |
| | FGANomaly | 1.46 | 0.18 | 5.97 | 0.93 | 0.82 | 1.13 | 2.44 | 2.84 | 0.72 | 1.02 |
| | TimesNet | 0.74 | 0.17 | 0.24 | 0.11 | 0.44 | 0.31 | 0.54 | 0.63 | 0.16 | 0.05 |
| | D3R | 2.11 | 0.6 | 3.84 | 0.72 | 0.72 | 0.33 | 1.19 | 1.67 | 7.17 | 1.56 |
| | SARAD | 1.60 | 0.38 | 1.33 | 0.20 | 0.56 | 0.51 | 0.79 | 1.09 | 0.48 | 1.11 |
| Diffusion Model | DiffAD | 0.85 | 0.42 | 0.17 | 0.08 | 0.34 | 0.39 | 0.74 | 0.45 | 0.2 | 0.03 |
| | **Ours** | 0.48 | 0.68 | 5.98 | 2.48 | 0.84 | 1.29 | 0.66 | 0.27 | 0.16 | 0.06 |

In addition to using precision, recall, and F1-score as evaluation metrics in Sections 5.2 and 5.3, we further employ a comprehensive set of measures including threshold-independent AUC-ROC and AUC-PR, completely parameter-free VUS-ROC and VUS-PR, the sequence-level evaluation metric Affiliation, and point-adjustment metrics for event-level evaluation, to enable a thorough assessment.

Table 5 reports the AUC-ROC and AUC-PR results of the evaluated methods, while Table 6 presents the corresponding standard deviations. Table 7 summarizes the VUS-ROC and VUS-PR scores, with their standard deviations provided in Table 8. Table 9 displays the Affiliation metric values, and Table 10 lists their respective standard deviations. Table 11 reports the point-adjustment metrics.

Experimental results demonstrate that ICDiffAD significantly outperforms prior diffusion-based approaches, achieving improvements of 23.45% in AUC-ROC, 18.65% in AUC-PR, 16.37% in VUS-ROC, 12.53% in VUS-PR and 8.93% in Affiliation-F1 over DiffAD (ImDiffusion, which relies on multi-step reconstruction and voting, is not compatible with these metrics). These findings further validate the effectiveness of our method in addressing the reconstruction instability inherent in vanilla diffusion models for TSAD.

Although ICDiffAD does not consistently achieve the best performance across all datasets, diffusion-based methods with input-consistent reconstruction exhibit strong overall performance and achieve high average scores. This underscores the considerable potential of diffusion models in TSAD, warranting further exploration.

### 8.4.3 STANDARD DEVIATIONS OF DETECTION PERFORMANCE

Table 12 reports the standard deviations of anomaly detetion performance as reported in Table 1 in Section 5.2.

Table 7: VUS-ROC and VUS-PR scores across five real-world datasets, where higher values indicate better performance. All values are average percentages from five random seeds. The best results are shown in **bold**, and the second-best results are underlined.

| Dataset | Method | MSL | | SMAP | | SMD | | PSM | | SWAT | | Average | |
|---|---|---|---|---|---|---|---|---|---|---|---|---|---|
| | | VUS-ROC | VUS-PR | VUS-ROC | VUS-PR | VUS-ROC | VUS-PR | VUS-ROC | VUS-PR | VUS-ROC | VUS-PR | VUS-ROC | VUS-PR |
| Classical | IF | 66.62 | 19.14 | **61.03** | **17.30** | 53.56 | 7.25 | 57.99 | 41.48 | 84.39 | 63.57 | 64.72 | 29.75 |
| | ECOD | 61.06 | 19.74 | 43.27 | 11.91 | 60.85 | 8.31 | 60.25 | 38.62 | 82.67 | 61.42 | 61.62 | 28.00 |
| | CBLOF | 67.94 | 22.37 | 57.12 | 17.01 | 65.24 | 9.97 | **64.80** | **46.25** | 78.38 | 52.99 | 66.70 | 29.72 |
| | PCA | 58.99 | 18.43 | 44.83 | 12.11 | 65.83 | 10.05 | 58.22 | 42.67 | 69.43 | 51.43 | 59.46 | 26.94 |
| Representation | Deep SVDD | 60.54 | 19.07 | 44.80 | 12.13 | 64.27 | 9.50 | 61.34 | 43.40 | 68.06 | 50.05 | 59.80 | 26.83 |
| | DCDetector | 53.87 | 15.58 | 51.61 | 14.78 | 48.75 | 4.40 | 43.66 | 25.06 | 50.76 | 14.37 | 49.73 | 14.84 |
| Recontruction | ATF-UAD | 61.63 | 19.06 | 43.44 | 11.59 | 43.10 | 4.97 | 46.03 | 33.18 | 54.35 | 20.58 | 49.71 | 17.88 |
| | AT | 51.05 | 14.45 | 52.02 | 15.99 | 49.97 | 5.36 | 36.82 | 26.47 | 46.45 | 12.79 | 47.26 | 15.01 |
| | FGANomaly | 57.51 | 15.39 | 48.61 | 12.95 | 70.27 | 9.19 | 58.94 | 45.60 | 83.90 | 66.61 | 63.85 | 29.95 |
| | TimesNet | 62.50 | 17.79 | 40.01 | 12.54 | 77.44 | **15.53** | 60.65 | 40.55 | 26.11 | 9.27 | 53.34 | 19.14 |
| | D3R | 64.92 | 20.57 | 48.97 | 13.45 | 71.39 | 14.45 | 57.50 | 37.01 | 59.22 | 17.08 | 60.40 | 20.51 |
| | SARAD | 62.92 | 21.69 | 42.71 | 12.83 | **79.67** | 15.02 | 61.77 | 41.01 | **87.52** | **70.68** | 66.92 | **32.25** |
| Diffusion Model | DiffAD | 60.25 | 18.79 | 56.80 | 15.45 | 58.4 | 7.19 | 51.02 | 32.01 | 26.45 | 9.21 | 50.58 | 16.53 |
| | **Ours** | **75.84** | **31.14** | 58.23 | 15.55 | 73.78 | 14.47 | 64.65 | 41.60 | 62.26 | 42.53 | **66.95** | 29.06 |

Table 8: Standard deviations of anomaly detection performance in Table 7. Standard deviations of VUS-ROC and VUS-PR scores are reported. All values are percentages.

| Dataset | Method | MSL | | SMAP | | SMD | | PSM | | SWAT | |
|---|---|---|---|---|---|---|---|---|---|---|---|
| | | VUS-ROC | VUS-PR | VUS-ROC | VUS-PR | VUS-ROC | VUS-PR | VUS-ROC | VUS-PR | VUS-ROC | VUS-PR |
| Classical | IF | 0.78 | 0.32 | 0.70 | 0.33 | 0.40 | 0.11 | 1.13 | 1.10 | 1.56 | 2.58 |
| | ECOD | 0.00 | 0.00 | 0.00 | 0.00 | 0.00 | 0.00 | 0.00 | 0.00 | 0.00 | 0.00 |
| | CBLOF | 1.30 | 0.81 | 0.28 | 0.19 | 1.32 | 0.19 | 0.42 | 0.77 | 0.99 | 0.95 |
| | PCA | 0.00 | 0.00 | 0.00 | 0.00 | 0.00 | 0.00 | 0.00 | 0.00 | 0.00 | 0.00 |
| Representation | Deep SVDD | 1.37 | 0.45 | 0.98 | 0.27 | 1.84 | 0.82 | 4.31 | 4.73 | 3.91 | 6.00 |
| | DCDetector | 2.39 | 0.16 | 1.83 | 0.43 | 0.39 | 0.12 | 1.57 | 0.13 | 0.06 | 0.24 |
| Recontruction | ATF-UAD | 3.26 | 1.92 | 4.34 | 2.72 | 2.19 | 0.12 | 1.33 | 1.74 | 11.92 | 6.27 |
| | AT | 1.21 | 1.35 | 1.52 | 1.23 | 0.39 | 0.48 | 3.22 | 0.86 | 3.61 | 0.43 |
| | FGANomaly | 1.14 | 0.18 | 5.35 | 0.95 | 0.73 | 0.38 | 2.29 | 1.83 | 1.20 | 3.37 |
| | TimesNet | 0.59 | 0.23 | 1.17 | 0.22 | 0.50 | 0.37 | 0.86 | 0.93 | 0.34 | 0.48 |
| | D3R | 1.77 | 0.73 | 1.82 | 0.44 | 2.46 | 0.37 | 1.43 | 1.77 | 6.38 | 1.68 |
| | SARAD | 1.60 | 0.60 | 1.43 | 0.24 | 1.01 | 0.76 | 1.09 | 0.69 | 0.60 | 1.05 |
| Diffusion Model | DiffAD | 1.07 | 0.59 | 0.20 | 0.08 | 0.31 | 0.18 | 0.89 | 0.4 | 0.3 | 0.12 |
| | **Ours** | 0.58 | 0.35 | 0.34 | 0.10 | 2.80 | 1.74 | 1.60 | 0.45 | 0.06 | 0.13 |

### 8.4.4 MODEL COMPLEXITY

We conduct a comprehensive comparison of neural network-based models on the SMD dataset in terms of training time, inference time, model size, and computational cost, to evaluate the practicality of ICDiffAD in real-world deployment scenarios. The results are presented in Table 13. All experiments related to time overhead were conducted on a single computing node equipped with an AMD EPYC 7443 CPU (24 cores, 96 threads), an NVIDIA 3090 GPU (24GB), and 256 GB of RAM.

As observed, due to the iterative nature of solving each diffusion step, diffusion-based models inherently require longer training and inference times. However, given that both the training and testing sets of SMD contain approximately 16 days of data, the additional overhead introduced by the diffusion process remains acceptable for practical deployment and maintenance. Moreover, thanks to the use of an SNR-based scheduler for optimized noise scheduling and the input-consistent reconstruction guided by SIC mechanism of input sequences, ICDiffAD achieves faster training and inference speeds compared to existing diffusion-based approaches.

### 8.4.5 ABLATION STUDY FOR DENOISING STEP

We conducted comprehensive experiments on five real-world datasets by testing fixed denoising steps during the reverse diffusion process. As shown in Table 14, the optimal denoising steps vary across different datasets, and all fixed-step configurations underperform compared to our SIC method. These results confirm that our proposed SIC approach effectively estimates the optimal corruption factor and denoising steps for various datasets.

### 8.5 BROADER IMPACTS

The societal implications of ICDiffAD extend beyond technical advancements in time series analysis. From a positive perspective, the method's ability to detect subtle anomalies in noisy, heterogeneous data could enhance predictive maintenance in critical infrastructure (e.g., power grids, transportation systems), potentially preventing catastrophic failures and reducing economic losses. In healthcare, improved anomaly detection in physiological time series (e.g., EEG, ECG) may enable earlier diagnosis of pathological conditions, though rigorous clinical validation would be required before

Table 9: Affiliation-based metrics across five real-world datasets, where higher values indicate better performance. All values are average percentages from five random seeds. The best results are shown in **bold**, and the second-best results are underlined.

| Dataset | Method | MSL | | | SMAP | | | SMD | | | PSM | | | SWaT | | | Average |
|---|---|---|---|---|---|---|---|---|---|---|---|---|---|---|---|---|---|
| | | Aff-P | Aff-R | Aff-F1 | Aff-P | Aff-R | Aff-F1 | Aff-P | Aff-R | Aff-F1 | Aff-P | Aff-R | Aff-F1 | Aff-P | Aff-R | Aff-F1 | Aff-F1 |
| Classical | IForst | 53.87 | 94.58 | 68.64 | 41.12 | 68.91 | 51.51 | 62.94 | 94.27 | 75.48 | 58.66 | 88.79 | 70.65 | 53.03 | 99.95 | 69.29 | 67.11 |
| | ECOD | 60.67 | 51.72 | 55.83 | 44.98 | 100.00 | 62.05 | 61.49 | 58.72 | 60.08 | 57.75 | 96.33 | 72.21 | 52.00 | 100.00 | 68.42 | 63.72 |
| | CBLOF | 58.94 | 90.69 | 71.45 | 42.41 | 100.00 | 59.56 | 63.51 | 62.33 | 62.92 | 60.02 | 100.00 | 75.02 | 60.64 | 100.00 | 75.50 | 68.89 |
| | PCA | 59.27 | 64.53 | 61.79 | 57.32 | 99.71 | **72.79** | 72.58 | 55.42 | 62.85 | 66.24 | 100.00 | **79.69** | 53.91 | 99.99 | 70.05 | 69.44 |
| Representation | Deep SVDD | 49.88 | 98.87 | 66.31 | 42.67 | 68.23 | 52.50 | 65.82 | 80.49 | 72.42 | 58.32 | 60.11 | 59.20 | 55.73 | 97.34 | 70.88 | 64.26 |
| | DCDetector | 55.94 | 95.53 | 70.56 | 45.92 | 97.49 | 62.43 | 50.93 | 93.57 | 65.96 | 54.72 | 83.36 | 66.07 | 53.25 | 98.12 | 69.03 | 66.81 |
| Recontruction | ATF-UAD | 74.00 | 95.81 | 83.50 | 44.98 | 100.00 | 62.05 | 48.10 | 99.17 | 64.78 | 58.41 | 91.58 | 71.33 | 60.08 | 99.71 | 74.98 | 71.33 |
| | AT | 51.04 | 95.36 | 66.49 | 52.91 | 96.69 | 68.39 | 54.08 | 97.07 | 69.46 | 54.26 | 82.18 | 65.36 | 53.63 | 98.27 | 69.39 | 67.82 |
| | FGANomaly | 56.36 | 98.27 | 71.64 | 50.44 | 97.62 | 66.51 | 53.46 | 83.25 | 65.11 | 55.31 | 98.34 | 70.80 | 59.93 | 81.52 | 69.08 | 68.63 |
| | TimesNet | 61.25 | 98.13 | 75.42 | 44.62 | 100.00 | 61.71 | 64.81 | 95.32 | 77.16 | 58.32 | 94.33 | 72.08 | 55.19 | 100.00 | 71.13 | 71.50 |
| | D3R | 66.85 | 90.83 | 77.02 | 51.76 | 92.55 | 66.39 | 60.87 | 97.93 | 75.08 | 63.32 | 88.721 | 73.90 | 60.14 | 97.57 | 74.41 | 73.36 |
| | SARAD | 56.90 | 99.54 | 72.41 | 47.94 | 100.00 | 64.81 | 59.00 | 99.99 | 74.21 | 47.14 | 99.88 | 64.05 | 59.28 | 100.00 | 74.43 | 69.98 |
| Diffusion Model | DiffAD | 65.56 | 100.00 | 79.20 | 57.34 | 61.46 | 59.33 | 53.23 | 91.17 | 67.21 | 54.58 | 84.17 | 66.22 | 62.11 | 100.00 | 76.63 | 69.72 |
| | **Ours** | 72.53 | 99.80 | **84.01** | 55.18 | 98.72 | 70.79 | 66.66 | 94.61 | **78.21** | 64.55 | 99.63 | 78.34 | 69.40 | 99.96 | **81.92** | **78.65** |

Table 10: Standard deviations of anomaly detection performance in Table 9. Standard deviations of Affiliation-based metrics are reported. All values are percentages.

| Dataset | Method | MSL | | | SMAP | | | SMD | | | PSM | | | SWaT | | |
|---|---|---|---|---|---|---|---|---|---|---|---|---|---|---|---|---|
| | | Aff-P | Aff-R | Aff-F1 | Aff-P | Aff-R | Aff-F1 | Aff-P | Aff-R | Aff-F1 | Aff-P | Aff-R | Aff-F1 | Aff-P | Aff-R | Aff-F1 |
| Classical | IForst | 0.67 | 3.21 | 0.98 | 0.77 | 2.86 | 0.74 | 1.05 | 1.78 | 0.44 | 1.91 | 4.63 | 1.01 | 0.11 | 0.07 | 0.05 |
| | ECOD | 0.00 | 0.00 | 0.00 | 0.00 | 0.00 | 0.00 | 0.00 | 0.00 | 0.00 | 0.00 | 0.00 | 0.00 | 0.00 | 0.00 | 0.00 |
| | CBLOF | 0.96 | 3.64 | 0.88 | 0.32 | 1.57 | 0.49 | 0.33 | 1.15 | 0.39 | 0.32 | 1.73 | 0.62 | 0.03 | 0.02 | 0.01 |
| | PCA | 0.00 | 0.00 | 0.00 | 0.00 | 0.00 | 0.00 | 0.00 | 0.00 | 0.00 | 0.00 | 0.00 | 0.00 | 0.00 | 0.00 | 0.00 |
| Representation | Deep SVDD | 1.07 | 2.67 | 1.33 | 0.18 | 2.91 | 0.34 | 1.41 | 0.77 | 0.82 | 3.17 | 5.92 | 3.77 | 1.00 | 0.37 | 0.33 |
| | DCDetector | 0.00 | 0.00 | 0.00 | 0.00 | 0.00 | 0.00 | 0.00 | 0.00 | 0.00 | 0.00 | 0.00 | 0.00 | 0.00 | 0.00 | 0.00 |
| Recontruction | ATF-UAD | 1.02 | 6.35 | 0.41 | 1.38 | 3.36 | 1.67 | 1.18 | 3.28 | 1.53 | 1.87 | 5.28 | 2.07 | 7.32 | 5.28 | 6.93 |
| | AT | 0.00 | 0.00 | 0.00 | 0.00 | 0.00 | 0.00 | 0.00 | 0.00 | 0.00 | 0.00 | 0.00 | 0.00 | 0.00 | 0.00 | 0.00 |
| | FGANomaly | 0.21 | 4.32 | 0.21 | 0.87 | 4.39 | 1.31 | 1.52 | 6.61 | 0.55 | 2.61 | 3.38 | 1.92 | 1.39 | 0.88 | 0.45 |
| | TimesNet | 0.30 | 6.46 | 0.38 | 0.16 | 0.85 | 0.26 | 0.49 | 2.69 | 0.44 | 0.98 | 7.76 | 0.20 | 0.00 | 0.00 | 0.00 |
| | D3R | 1.31 | 3.67 | 0.99 | 0.24 | 4.28 | 0.33 | 0.29 | 1.32 | 2.14 | 0.48 | 2.98 | 0.10 | 1.49 | 4.38 | 1.27 |
| | SARAD | 0.81 | 10.32 | 0.43 | 0.28 | 1.99 | 0.33 | 0.88 | 4.34 | 1.28 | 1.33 | 0.87 | 1.52 | 1.09 | 1.71 | 0.42 |
| Diffusion Model | DiffAD | 0.02 | 1.24 | 0.02 | 0.06 | 1.38 | 0.04 | 0.22 | 0.78 | 0.36 | 0.00 | 0.01 | 0.00 | 0.00 | 0.00 | 0.00 |
| | **Ours** | 2.44 | 3.39 | 0.64 | 0.12 | 0.51 | 0.23 | 1.32 | 2.28 | 0.11 | 0.19 | 1.74 | 0.08 | 0.00 | 0.00 | 0.00 |

deployment. Methodologically, our SNR-centric framework advances the integration of diffusion models into discriminative tasks, offering insights into balancing stochastic generation with input-consistent reconstruction in generative AI applications such as medical imaging and climate modeling.

Table 11: Point-adjustment metrics across five real-world datasets, where higher values indicate better performance. All values are average percentages from five random seeds. The best results are shown in **bold**, and the second-best results are underlined.

| Dataset | | MSL | | | SMAP | | | SMD | | | PSM | | | SWaT | | | Average |
|---|---|---|---|---|---|---|---|---|---|---|---|---|---|---|---|---|---|---|
| Method | | P | R | F1 | P | R | F1 | P | R | F1 | P | R | F1 | P | R | F1 | F1 |
| Classical | IForst | 53.94 | 86.54 | 66.45 | 52.39 | 59.07 | 55.53 | 42.31 | 73.29 | 53.64 | 76.09 | 92.45 | 83.48 | 49.29 | 44.95 | 47.02 | 61.22 |
| | ECOD | 66.14 | 73.18 | 69.48 | 71.41 | 76.33 | 73.79 | 86.45 | 75.73 | 80.74 | 75.65 | 86.32 | 80.63 | 69.54 | 68.32 | 68.92 | 74.71 |
| | CBLOF | 59.38 | 69.91 | 64.22 | 80.31 | 71.39 | 75.59 | 79.70 | 76.98 | 78.32 | 83.01 | 90.96 | 86.80 | 71.35 | 80.29 | 75.56 | 76.10 |
| | PCA | 62.18 | 77.58 | 69.03 | 76.39 | 62.19 | 68.56 | 55.64 | 69.18 | 61.68 | 81.92 | 91.28 | 86.35 | 59.28 | 77.29 | 67.10 | 70.54 |
| Representation | Deep SVDD | 91.92 | 76.63 | 83.58 | 89.93 | 56.02 | 69.04 | 78.54 | 79.67 | 79.10 | 95.41 | 86.49 | 90.73 | 80.42 | 84.45 | 82.39 | 80.97 |
| | DCDetector | 93.69 | 99.69 | **96.60** | 95.63 | 98.92 | 97.25 | 83.59 | 91.10 | 87.18 | 97.14 | 98.74 | 97.93 | 93.11 | 99.77 | 96.33 | 95.06 |
| Recontruction | ATF-UAD | 90.18 | 94.34 | 92.21 | 92.82 | 96.49 | 94.62 | 88.29 | 92.37 | 90.28 | 95.93 | 94.18 | 95.05 | 93.19 | 97.28 | 95.19 | 93.47 |
| | AT | 92.09 | 95.15 | 93.59 | 94.13 | 99.40 | 96.69 | 89.40 | 95.45 | 92.33 | 96.91 | 98.90 | 97.89 | 91.55 | 96.73 | 94.07 | 94.92 |
| | FGANomaly | 87.58 | 90.29 | 88.91 | 93.34 | 90.27 | 91.78 | 85.63 | 89.27 | 87.41 | 95.34 | 97.52 | 96.42 | 88.74 | 90.98 | 89.85 | 90.87 |
| | TimesNet | 83.92 | 86.42 | 85.15 | 92.52 | 58.29 | 71.52 | 88.66 | 83.14 | 85.81 | 98.19 | 96.76 | 97.47 | 86.76 | 97.32 | 91.74 | 86.34 |
| | D3R | 88.19 | 87.09 | 87.64 | 86.37 | 89.09 | 87.71 | 87.74 | 96.49 | 91.91 | 93.84 | 99.21 | 96.45 | 83.09 | 83.00 | 83.04 | 89.35 |
| | SARAD | 91.28 | 93.59 | 92.42 | 90.28 | 94.13 | 92.16 | 88.14 | 87.39 | 87.76 | 95.37 | 95.27 | 95.32 | 92.46 | 96.33 | 94.36 | 92.40 |
| Diffusion Model | DiffAD | 92.97 | 95.44 | 94.19 | 96.52 | 97.38 | 96.95 | 90.01 | 95.67 | 92.75 | 97.00 | 98.92 | 97.95 | 98.44 | 96.90 | 97.66 | 95.90 |
| | ImDiffusion | 89.30 | 86.38 | 87.82 | 87.71 | 96.18 | 91.75 | 95.20 | 95.09 | 95.14 | 98.11 | 97.53 | 97.82 | 89.88 | 84.65 | 87.19 | 91.94 |
| | Ours | 95.29 | 94.32 | 94.80 | 98.11 | 97.09 | **97.60** | 95.81 | 95.50 | **95.65** | 98.02 | 98.61 | **98.31** | 97.19 | 99.01 | **98.09** | **96.89** |

Table 12: Standard deviations of anomaly detection performance in Table 1. Standard deviations of threshold-dependent Precision, Recall, F1 scores are reported. All values are percentages.

| Dataset | | MSL | | | SMAP | | | SMD | | | PSM | | | SWaT | | |
|---|---|---|---|---|---|---|---|---|---|---|---|---|---|---|---|---|
| Method | | P | R | F1 | P | R | F1 | P | R | F1 | P | R | F1 | P | R | F1 |
| Classical | IF | 1.17 | 6.49 | 1.47 | 0.71 | 2.99 | 0.76 | 1.17 | 2.08 | 0.51 | 2.79 | 10.28 | 1.39 | 0.22 | 0.11 | 0.03 |
| | ECOD | 0.00 | 0.00 | 0.00 | 0.00 | 0.00 | 0.00 | 0.00 | 0.00 | 0.00 | 0.00 | 0.00 | 0.00 | 0.00 | 0.00 | 0.00 |
| | CBLOF | 1.38 | 9.22 | 1.13 | 0.44 | 4.16 | 0.95 | 0.45 | 1.60 | 0.58 | 0.44 | 4.09 | 1.06 | 0.03 | 0.02 | 0.01 |
| | PCA | 0.00 | 0.00 | 0.00 | 0.00 | 0.00 | 0.00 | 0.00 | 0.00 | 0.00 | 0.00 | 0.00 | 0.00 | 0.00 | 0.00 | 0.00 |
| Representation | Deep SVDD | 2.06 | 7.72 | 1.94 | 0.20 | 6.16 | 0.21 | 2.72 | 1.55 | 1.81 | 5.49 | 13.84 | 3.64 | 1.00 | 0.37 | 0.33 |
| | DCDetector | 0.00 | 0.00 | 0.00 | 0.00 | 0.00 | 0.00 | 0.00 | 0.00 | 0.00 | 0.00 | 0.00 | 0.00 | 0.00 | 0.00 | 0.00 |
| Recontruction | ATF-UAD | 1.59 | 24.88 | 0.46 | 1.58 | 0.01 | 1.07 | 1.27 | 7.19 | 1.67 | 2.88 | 9.71 | 2.36 | 13.61 | 9.88 | 12.34 |
| | AT | 0.00 | 0.00 | 0.00 | 0.00 | 0.00 | 0.00 | 0.00 | 0.00 | 0.00 | 0.00 | 0.00 | 0.00 | 0.00 | 0.00 | 0.00 |
| | FGANomaly | 0.25 | 0.62 | 0.20 | 1.98 | 0.00 | 1.37 | 1.96 | 14.00 | 0.64 | 3.78 | 7.56 | 2.51 | 1.78 | 0.86 | 0.42 |
| | TimesNet | 0.30 | 6.46 | 0.38 | 0.16 | 0.85 | 0.26 | 0.49 | 2.69 | 0.44 | 0.98 | 0.00 | 0.20 | 0.00 | 0.00 | 0.00 |
| | D3R | 1.13 | 7.76 | 1.36 | 0.25 | 2.78 | 0.12 | 0.31 | 1.58 | 4.99 | 0.57 | 5.46 | 0.11 | 2.58 | 8.51 | 3.27 |
| | SARAD | 0.70 | 8.08 | 0.47 | 0.19 | 0.00 | 0.10 | 0.76 | 4.92 | 1.38 | 1.65 | 0.65 | 1.06 | 1.47 | 1.39 | 0.88 |
| Diffusion Model | DiffAD | 0.02 | 0.00 | 0.02 | 0.06 | 0.01 | 0.04 | 0.38 | 0.66 | 0.32 | 0.00 | 0.01 | 0.00 | 0.00 | 0.00 | 0.00 |
| | ImDiffusion | 2.85 | 0.45 | 0.82 | 0.38 | 0.15 | 0.22 | 1.32 | 0.67 | 0.82 | 6.43 | 0.47 | 0.87 | NA | NA | NA |
| | **Ours** | 3.00 | 9.19 | 0.74 | 0.12 | 0.62 | 0.21 | 2.15 | 3.83 | 0.17 | 0.20 | 2.45 | 0.09 | 0.00 | 0.00 | 0.00 |

Table 13: Model complexity and overheads. The total training time (in seconds), the total inference time (in seconds), the total number of parameters (in MB) and the GFLOPS(Giga Floating Point Operations Per Second) are reported.

| Method | Training Time (s) | Inference Time (s) | Total Params (MB) | GFLOPS |
|---|---|---|---|---|
| Deep SVDD | 43.73 | 0.25 | 0.02 | 0.01 |
| DCDetector | 1522.39 | 337.61 | 0.66 | 198.10 |
| ATF-UAD | 6922.81 | 239.81 | 0.14 | 24.10 |
| AT | 19.88 | 4.22 | 4.83 | 123.47 |
| FGANomaly | 1678.43 | 13.82 | 0.09 | 0.01 |
| TimesNet | 144.02 | 12.20 | 4.70 | 319.22 |
| D3R | 8960.97 | 1520.63 | 52.24 | 28.62 |
| SARAD | 808.20 | 71.00 | 9.55 | 92.45 |
| DiffAD | 3299.53 | 2468.00 | 38.83 | 310.29 |
| ImDiffusion | 62003.90 | 3043.42 | 0.31 | 51.84 |
| **Ours** | 1282.85 | 166.34 | 30.82 | 716.96 |

Table 14: Ablation study for denoising step. The standard F1 scores are reported.

| Denoising Step | MSL | SMAP | SMD | PSM | SWaT |
|---|---|---|---|---|---|
| 90% | 24.33 | 23.94 | 19.88 | 48.91 | 76.51 |
| 70% | 29.12 | 25.01 | 21.32 | 52.37 | 77.08 |
| 50% | 32.60 | 26.27 | 22.84 | 54.80 | 77.19 |
| 30% | 31.19 | 27.33 | 22.71 | 54.83 | 76.87 |
| 10% | 28.71 | 25.98 | 20.33 | 49.61 | 76.33 |

