# OpenReview forum: "ICDiffAD: Implicit Conditioning Diffusion Model for Time Series Anomaly Detection"
_ICLR.cc/2026/Conference — ICLR 2026 Poster_

### Official Review · Reviewer_uRQA · 2025-10-17

**Soundness:** 2
**Presentation:** 1
**Contribution:** 2
**Rating:** 2
**Confidence:** 3

**Summary:**

This paper aims to address the reconstruction inconsistency problem in diffusion-based time series anomaly detection, which arises from the noise-adding and denoising processes. Specifically, the authors argue that when a sample is diffused all the way to pure noise, the strong generative capability of diffusion models can produce highly diverse reconstructions, thereby degrading anomaly detection performance. The paper focuses on optimizing the noise scheduler during both training and testing. In the training stage, the authors replace conventional noise scheduling strategies (e.g., cosine schedule) with one based on the signal-to-noise ratio (SNR). During inference, they further adapt the noise strength per sample, again guided by the SNR. Experimental results across multiple datasets and various baselines demonstrate that the proposed method performs among the top tier. In particular, compared with other diffusion-based approaches, the proposed technique shows improvement in efficiency.

**Strengths:**

The paper has two notable merits, summarized as follows:

1.  The paper introduces a novel perspective by optimizing anomaly detection from the viewpoint of noise scheduling. In my opinion, this is an interesting and original angle, and to some extent represents a new direction for improving both the accuracy and efficiency of diffusion models for conditional time series generation.
2.  The experimental evaluation is reasonably comprehensive. In particular, the ablation studies are well designed and clearly demonstrate the effectiveness of each component in the proposed framework.

**Weaknesses:**

Similarly, from my perspective, the current version of this paper also exhibits several non-negligible weaknesses, summarized as follows:

1. **Motivation.**
   First, I acknowledge that generative models can produce diverse outputs when starting from different noise priors. I also notice that the paper (see Fig. 1) implicitly assumes that the denoised samples obtained from diffusion models correspond to “normal” (i.e., smooth) samples. This assumption itself is reasonable. However, later in the paper (lines 278–279), the authors state their own premise—*“the premise that time series anomalies manifest as high-frequency deviations from normative patterns.”*
   Given this assumption, one could argue that existing diffusion-based methods are already capable of separating such high-frequency anomalies from normal patterns. Furthermore, regarding Fig. 1, I am skeptical about the second-row examples: such patterns could naturally arise from any sliding-window training process, and thus may not genuinely indicate anomalous behavior.

2. **Writing and presentation.**
   The narrative flow of the paper is not particularly convincing. For instance, the introduction of the signal-to-noise ratio (SNR) is abrupt and lacks conceptual grounding, despite being the core of the entire framework. Merely pointing out the potential limitations of previous scheduling methods is insufficient; the authors should elaborate on how SNR-based scheduling specifically benefits time-series data.
   In addition, Fig. 2 provides only a coarse overview. A more detailed modular diagram or code block would be valuable for understanding the system design.

3. **Methodology.**
   The claim made in the *Introduction*—that *“this new scheduler induces the model to learn normal patterns across diverse, SNR-calibrated corruption levels, enhancing robustness to noise perturbations and temporal heterogeneity”*—is intriguing but not well supported. I am not convinced that a more linear ($\alpha$)-curve (as shown in Fig. 2) would directly yield these benefits.
   Moreover, the first module of the method seems fully general and could be applied to computer vision or other domains, with no specific connection to time-series data. In contrast, the second module, *Diffusion Step Alignment*, appears to disrupt the previously optimized noise-level alignment, which raises additional concerns.

4. **Experiments.**
   When reading the *Experiments* section, I was expecting to see visualizations of real-world data similar to those in Fig. 1, which would strongly support the paper’s central claims—but none are provided. The main experiments focus solely on point anomaly detection, with weak evaluation on localization tasks.

5. **Reproducibility.**
   Although the authors state that they plan to release the code in the future, the current submission does not provide sufficient implementation details to ensure reproducibility.

**Questions:**

1. See the Weaknesses part.
2. Are the reconstruction results in Fig. 2 produced by the proposed method? If so, the quality of the reconstructed samples appears suboptimal—could the authors clarify this?
3. In Fig. 3, the performance of DiffAD seems extremely poor, almost equivalent to random output. Could the authors explain the cause of this behavior or verify whether there might be an issue in the experimental configuration?
4. How is the inference time reported in Table 12 calculated? Does it correspond to a single forward pass of the neural network, or to the entire denoising process? The reported time seems unusually large.
5. Given that ImDiffusion has a relatively small number of parameters (also see Table 12), is the comparison fair? Have the authors normalized or controlled for model capacity when reporting efficiency results?

---

> ### Author Response · Authors · 2025-11-27
>
> | ID | Concern | Reply |
> | --- | --- | --- |
> | W1 | Motivation | Our hypothesis that "the premise that time series anomalies manifest as high-frequency deviations from normative patterns." employs the term "high-frequency anomalies" to describe abrupt transitions from normal states. However, such high-frequency anomalies may not be reliably detectable in windowed time series. We will address this potential misunderstanding through necessary revisions in our subsequent submission. Additionally, the caption of Figure 1 explicitly states that the second row illustrates possible reconstruction results from the diffusion model due to non-deterministic generation, rather than representing anomalous signals. |
> | W2 | Writing and presentation | SNR is a fundamental concept closely related to time series data. The SNR scheduler progressively injects noise based on the amount of information retained in the input, using SNR as an objective metric. This ensures that the denoising network can perform more accurate noise reduction adapted to time series with different characteristics. The SNR implicit conditioning mechanism adaptively estimates the optimal corruption factor and denoising steps according to the input data, and introduces implicit conditioning to achieve consistent reconstruction of the original input. Furthermore, Figure 2 illustrates the complete workflow of our method. The detailed mechanisms of SNR-based noise scheduling and SNR implicit conditioning are thoroughly explained in Sections 4.3 and 4.4, respectively. |
> | W3 | Methodology | The ablation studies in Table 2 demonstrate the positive impact of the SNR scheduler on time series anomaly detection performance. Regarding the alpha curve, it is important to clarify that our method does not make it more linear. The actual formulation for alpha generation is explicitly provided in Equation (9) of our paper. The diffusion step alignment mechanism forms the second stage of our SNR implicit conditioning module. Its function is to select the appropriate noise-injection step based on the estimated optimal corruption factor. This selected intermediate step then serves as the starting point for the reverse denoising process, enabling input-consistent generation while maintaining reconstruction fidelity. |
> | W4 | Experiments | Figure 3 demonstrates the visual reconstruction results on real-world datasets, showing that our method achieves faithful input-consistent generation. It should be noted that in the field of time series anomaly detection, point anomaly detection shares the same objective as localization tasks in image-based anomaly detection. |
> | W5 | Reproducibility | We have not released the code at this stage, but the implementation details are provided in Section 7.3.3 of the paper. |
> | Q2 | Reconstruction Results in Figure 2 | Figure 2 presents reconstruction results generated by our proposed method using the SMD dataset, which represents a highly heterogeneous time series collection. Our method successfully reconstructs the overall trend of this dataset, whereas other diffusion-based approaches show notably poor reconstruction performance. Additionally, Figure 3 provides visual comparisons with other methods on another real-world dataset, further demonstrating our method's capability to achieve input-consistent generation. |
> | Q3 | Poor Reconstruction Performance of DiffAD | We strictly used the code and parameter configurations from the official DiffAD codebase. Under the point-adjusted evaluation protocol, we successfully reproduced the metrics reported in the original paper. The DiffAD visualization in Figure 3 also corresponds to reconstruction results obtained under the same experimental setup. |
> | Q4 | Inference Time | The inference time reported in Table 12 represents the total time required to process the entire SMD test set. For diffusion-based methods, this includes the complete denoising process. The extended inference time is attributable to the substantial size of the SMD dataset. Relevant dataset specifications are provided in Section 7.3.1. |
> | Q5 | Concerns Regarding Number of Parameters | We strictly followed the official ImDiffusion implementation. Unlike DiffAD and our method, which employ U-Net as the denoising network, ImDiffusion utilizes a transformer architecture. Our baseline experiments fully adhered to the original ImDiffusion paper design, and we successfully reproduced the reported results under point-adjusted evaluation. In Table 12, the parameter counts for diffusion-based methods refer specifically to their denoising networks, while other methods report their total model parameters. No normalization or controlled adjustment of model capacity was applied in these comparisons.
>  |

---

> > ### Comment · Reviewer_uRQA · 2025-11-27
> >
> > I appreciate the authors’ response. I strongly recommend that the authors highlight the revisions in the revised draft using different colors. In addition, since the authors have addressed some of my concerns, I have increased my score. However, the issues regarding the code and the narrative still remain unsatisfactory to me; therefore, my overall score remains negative.

---

> > > ### Author Response · Authors · 2025-11-30
> > >
> > > Thank you for your suggestions. We have highlighted the revisions in blue in the updated manuscript. Although our code is not yet publicly available, we have provided a detailed explanation of the mechanism underlying our method in Sections 4.2, 4.3, and 4.4, and included implementation details in Appendix Section 7.3.3 to ensure reproducibility. Moreover, since SNR is inherently and strongly connected to time series data, we further explain its significance in diffusion-based TSAD in Section 4.1, naturally introducing it into our method and elaborating on how the SNR-based scheduling strategy specifically benefits time series data.

---

### Official Review · Reviewer_SPCE · 2025-10-20

**Soundness:** 2
**Presentation:** 3
**Contribution:** 3
**Rating:** 4
**Confidence:** 4

**Summary:**

The paper proposes a diffusion-based method for time series anomaly detection called ICDiffAD. It has two main parts. First, an signal-to-noise ratio (SNR) guided scheduler that replaces standard $\beta$ schedules by targeting a terminal SNR and deriving per-step $\alpha_t$ from a function $g(t)$. This makes the corruption level interpretable and consistent across runs. Second, SNR Implicit Conditioning (SIC) that estimates an input SNR by splitting each test window into a low-pass signal and a residual, then maps this estimate to a target signal-retention $\alpha^\*$ and chooses the closest diffusion step $\hat T$. The reverse process then starts from $x_{\hat T}$ and reconstructs to $x_0$. Anomaly scores are the reconstruction residuals on each window. The model uses a UNet-style backbone on 2D reshaped windows and keeps one set of hyperparameters across datasets. Experiments on MSL, SMAP, SMD, PSM, and SWaT report higher point-level F1 than prior diffusion baselines and small gains over the strongest reconstruction baseline. Ablations indicate that both the SNR scheduler and SIC contribute to the improvements.

**Strengths:**

- The proposed SNR Implicit Conditioning (SIC) adaptively estimates an optimal corruption level and denoising step for each test instance. This mechanism effectively selects the optimal diffusion time $\hat T$ per input, balancing reconstruction fidelity and determinism.

- The reported improvements are consistent across multiple benchmarks. Ablation studies (TSNR sweeps, $g(t)$ robustness, SIC contributions) and qualitative visualizations substantiate the claims, showing that performance gains come from the model design.

- The authors employ point-level metrics instead of range-based ones, avoiding inflated results and better reflecting the intrinsic difficulty of time-series anomaly localization.

- The model uses a fixed set of core hyperparameters across datasets (e.g., $T = 200$, window size $= 1024$, fixed $\mathrm{TSNR_{dB}}$), which enhances reproducibility and ensures fair comparisons across benchmarks.

**Weaknesses:**

- The statement that input-agnostic schedulers “indiscriminately degrade all frequency components” and “over-corrupt transients” is asserted without citation or evidence. This statement is also unclear.

- In Sections 4.2 and 4.3, the variable $Z$ appears with both $l$ and $t$ subscripts, but it is unclear which one corresponds to $Z_0$ in expressions such as $\mathbb{E}\[\||Z_0\||_F^2\]$. Please clarify the intended meaning of each $Z$ symbol.

- The text suggests that $\alpha_t$ and $M_t$ depend on each other, which reads as circular. Please state explicitly that a target terminal TSNR is first chosen, a specific example would clarify this. Also fix notation in the $\mathrm{SNR}(t)$ equation (10) to use $\bar{\alpha}_t$ rather than $\bar{\alpha}_T$.

- The paper repeatedly claims to achieve *deterministic reconstruction*, but the proposed procedure still **samples noise** when initializing the reverse process at $\hat T$ via $x_{\hat T} = \sqrt{\bar\alpha_{\hat T}}x_0 + \sqrt{1-\bar\alpha_{\hat T}}\varepsilon.$ This is **not** determinism in the diffusion sense (which would correspond to DDIM with $\eta=0$ or a closed-form sampling path). The goal of the paper is to get **input-consistent reconstruction rather than strict determinism**, so the terminology is confusing/misleading.

- The paper reports only strict point-level metrics as the primary evaluation, but does not include point-adjusted or event-level scores used by many prior TSAD works. While strict evaluation is commendable and should remain the main protocol, adding point-adjusted or event-based metrics (even in the appendix) would allow fair comparison with earlier baselines that report only lenient metrics. Without both views, it is difficult to fully contextualize the reported gains.

- Efficiency is mixed up with speed rather than real compute cost. Table 12 shows higher GFLOPS than DiffAD and ImDiffusion but still lower runtime, which suggests the model is actually doing more work per input but runs faster thanks to better GPU utilization or code efficiency. It is therefore unclear whether the method is truly efficient or simply well-optimized.

- It is unclear whether the proposed *SNR Implicit Conditioning* actually achieves the intended effect. The paper does not report the distribution of selected start steps $\hat{T}$ (e.g., mean and standard deviation per dataset), so it is difficult to verify that the ISNR-based selection is meaningful. A simple baseline would be to perform reconstruction starting from a fixed reduced step (e.g. $T/2$) as a hyperparameter, instead of always starting from full corruption. Without such comparison, it is hard to tell whether the gains come from the conditioning mechanism itself or from simply using fewer denoising steps.

**Questions:**

- What do the superscripts in $x^{\text{tar}}$ and $x^{\text{con}}$ stand for? Please define the terms “tar” and “con” explicitly when first introduced to avoid ambiguity.

- Why is GFLOPS highest for your model while wall-clock is lowest among diffusion baselines?

- How does the proposed method compare to baseline in a point-adjusted evaluation protocol?

---

> ### Author Response · Authors · 2025-11-20
>
> | ID | Concern | Reply |
> | --- | --- | --- |
> | W1 | Unclear Statement | We apologize for the confusion. In our paper, “indiscriminately degrade all frequency components” is describing uniform noise injection across different time series pattern; “over-corrupt transients” is describing the noise injection is too severe such that one step noise injection might remove the noise signal entirely. In our experiment, we observe such case for current diffusion models (even with the suggested parameters given by the authors). Our solution, include a SNR scheduler to enhance the noise estimation capability of the denoising network under varying levels of noise injection. By implementing a more refined noise injection strategy that specifically regulates information retention, we achieve more precise noise estimation across different levels of information preservation. This establishes the foundation for the SNR-based implicit conditioning mechanism to guide semi-deterministic reconstruction of the input data. In the subsequence submission, we will update these statements for a clearer description of the problem. |
> | W2 | Confusing Notation | Thank you for pointing this out. In Section 4.2, the subscript *l* refers to the temporal dimension of the time series data, while in Section 4.3, the subscript *t* denotes the diffusion step index. We will clarify this distinction in the subsequent version to prevent any confusion. |
> | W3 | SNR Scheduler’s Parameters | Thank you for the suggestion. We will state clearly that TSNR is the input parameter. We use $M_t$ to denote the signal energy after $t$ steps of noise injection. The SNR first calculates the ratio of $M_T$ to $M_0$ based on the specified TSNR. Then, according to the defined temporal corruption scheduler, it generates the $\alpha_t$ sequence to complete the noise scheduling. Thank you for pointing this out. We will make the corresponding corrections in the subsequent version. |
> | W4 | Strict Deterministic Reconstruction  | Thank you for your comment. We would like to clarify that deterministic reconstruction of the input is a prerequisite for reconstruction-based time series anomaly detection methods. Our approach achieves this prerequisite through the semi-deterministic generation capability of diffusion models. These represent two distinct conceptual dimensions. |
> | W5,Q3 | Additional Metrics | Thank you for the suggestion. We will include point-adjusted metrics in the appendix. Additionally, besides point adjustment, the affiliation and VUS metrics can also be considered as event-level evaluation criteria, which we have already reported in the appendix. |
> | W6,Q2 | Mismatch between GFLOPS and Inference speed in results | DiffAD and ImDiffusion are imputation-based diffusion methods that require generating numerous masks during training and inference to occlude portions of time series data, whereas our approach is grounded in traditional diffusion-based generation. Although their GFLOPS are lower than ours, their actual training and inference time exceeds ours due to the computational overhead induced by extensive masking. Moreover, during inference, our method incorporates an SNR-implicit conditioning mechanism that adaptively estimates the optimal corruption factor and denoising steps, significantly accelerating inference speed. |
> | W7 | Effectiveness of SIC | In the third row of the ablation studies in Table 2 (without the SIC method), the starting step of reconstruction is set to $T/2$. This demonstrates that our proposed SIC method successfully estimates the optimal corruption factor and denoising steps. Furthermore, we will conduct experiments with a fixed set of reconstruction starting steps, and the results are presented in Table R3.1. We will include these experiments in the appendix of the subsequent version. |
> | Q1 | Confusing Notation | The superscripts of $x^{con}$ and $x^{tar} $denote the conditional and target parts, respectively. Following the conventional notation of previous diffusion-based methods' partitioning convention, the input data is partitioned into visible and invisible segments, utilizing the visible portion as the condition $x^{con}$ to facilitate the generation of the invisible portion $x^{tar}$. Thank you for your feedback. We will make the corresponding revisions in the subsequent version. |
>
> Table R3.1: F1 Score Comparing Different Denoising Steps
>
> | Percentage of Denoising Steps | MSL | SMAP | SMD | PSM | SWAT |
> | --- | --- | --- | --- | --- | --- |
> | 90% | 24.33 | 23.94 | 19.88 | 48.91 | 76.51 |
> | 70% | 29.12 | 25.01 | 21.32 | 52.37 | 77.08 |
> | 50% | 32.60 | 26.27 | 22.84 | 54.80 | 77.19 |
> | 30% | 31.19 | 27.33 | 22.71 | 54.83 | 76.87 |
> | 10% | 28.71 | 25.98 | 20.33 | 49.61 | 76.33 |

---

> > ### Comment · Reviewer_SPCE · 2025-11-26
> >
> > Thank you for your detailed response and the additional experiments.
> >
> > I have read the rebuttal and appreciate the clarifications provided.
> >
> > **[W2, W3, Q1]** Thank you for the clarification regarding notation and definitions. I believe these have been fixed in the current revision.
> >
> > **[W5, Q3]** The paper has been updated with the new point-adjusted metrics table, which helps contextualize the performance against prior work that used lenient metrics.
> >
> > **[W6]** The explanation regarding the trade-off between mask generation overhead in baselines vs. your method's inference speed is clear.
> >
> > **[W7]** The additional results in Table R3.1 are convincing and demonstrate that the adaptive step selection outperforms fixed-step heuristics. Please ensure this table is included in the final appendix.
> >
> > However, for **W1** and **W4**, I have supplemental comments and strong suggestions for the final manuscript:
> >
> > **[W1]** I maintain that the current framing in the paper is not fully convincing:
> >
> > The rebuttal states that fixed schedulers "over-corrupt transients," but this is an observation specific to certain hyperparameter settings rather than a fundamental theoretical flaw of fixed schedulers (which could, conversely, under-corrupt if tuned differently). Asserting that standard schedulers "indiscriminately degrade" components without citation or theoretical proof weakens the argument.
> >
> > I strongly suggest reframing the motivation in the final version. Instead of critiquing standard schedulers as inherently "ill-posed," you should pivot to highlighting the positive attributes of your method. The strength of your approach lies in its input-awareness and the flexibility of the SNR-based parameterization, which guarantees consistent signal energy retention regardless of the dataset characteristics. Framing the contribution around adaptability rather than correction of a baseline flaw would be much more compelling and scientifically accurate.
> >
> > **[W4]** I acknowledge your point that the goal is stable reconstruction for anomaly detection, and I note that you use the term "semi-deterministic" to nuance this claim. However, "semi-deterministic" is somewhat vague and potentially confusing in the diffusion literature, where "deterministic" implies specific sampling paths (e.g., ODE samplers). Since your method still relies on sampling stochastic noise ϵ, it is mathematically stochastic.
> >
> > More importantly, the term "semi-deterministic" undersells the actual mechanism of your approach. The strength of your method isn't that it is "half-deterministic," but rather that it is "input-consistent." By initializing the reverse process at $\hat{T}$ (a partially corrupted version of the input) rather than T (pure isotropic noise), you are constraining the generative trajectory to remain close to the input manifold. This constraint reduces variance and ensures fidelity, which is the key requirement for TSAD.
> >
> > I strongly suggest adopting terms like "input-consistent" or "stable reconstruction" in the final manuscript. This would not only resolve the terminological inaccuracy but also more precisely frame how your method solves the reconstruction instability problem
> >
> > **Conclusion**
> >
> > Based on the clarifications and the solid results provided in the rebuttal (specifically the robustness shown in Table R3.1), I am inclined to view the paper more favorably. Could the authors specifically summarize what concrete changes they have made (vs will make) in the current manuscript?

---

> > > ### Author Response · Authors · 2025-11-27
> > >
> > > Thank you for your thoughtful review of our work and the effort invested in evaluating our submission. We fully agree with your comments regarding W1 and W4. Directly emphasizing the adaptability of the SNR scheduler and adopting the term "input-consistent" will indeed make our framework more compelling.
> > >
> > > The current revision includes the following key updates:
> > >
> > > 1.Expanded the survey of diffusion-based time series anomaly detection methods in the Related Work section.
> > >
> > > 2.Corrected notations in Equations (6) and (10), clarified symbols in Section 3.2, and explicitly stated that the TSNR in Section 4.3 is user-specified.
> > >
> > > 3.Revised the description in Section 4.1 from "indiscriminately reducing all frequency components" to "uniformly injecting noise across different time series patterns."
> > >
> > > 4.Updated Section 4.4 to remove the earlier assumption that “normal sequences have lower frequencies and anomalous sequences higher frequencies,” and clarified that the SIC mechanism determines the optimal denoising steps and corruption factor through an adaptive assessment of input data complexity, independent of anomalous-sequence frequency.
> > >
> > > 5.Added point-adjusted (event-level) evaluation in Appendix 7.4.2.
> > >
> > > 6.Included ablation studies on fixed denoising steps in Appendix 7.4.5.
> > >
> > > For the next revision, we plan to:
> > >
> > > 1.Reframe the motivation of the SNR scheduler, highlighting its adaptive nature to input data as a key contribution.
> > >
> > > 2.Replace all instances of the term “semi-deterministic generation” with “input-consistent generation” and revise the manuscript accordingly.

---

> > > > ### Comment · Reviewer_SPCE · 2025-11-27
> > > >
> > > > Thank you for the summary.
> > > >
> > > > To clarify, if I see **W1** and **W4** addressed in the revised manuscript, I would be willing to raise my score again. Please keep in mind that you may use a supplementary (10th) page for the revised manuscript, in accordance with the ICLR guidelines.

---

> > > > > ### Author Response · Authors · 2025-11-30
> > > > >
> > > > > Thank you for your prompt response and recognition of our research. Regarding W1, we have rewritten Section 4.1 in the revised manuscript to clarify the motivation of the SNR scheduler and to emphasize that our contribution lies in its adaptability to the input data. Regarding W2, we have replaced the term “semi-deterministic generation” with “input-consistent generation” and updated other related terminology accordingly.

---

### Official Review · Reviewer_J9Mc · 2025-10-28

**Soundness:** 2
**Presentation:** 2
**Contribution:** 2
**Rating:** 4
**Confidence:** 4

**Summary:**

This paper proposes ICDiffAD, a diffusion-based framework for time-series anomaly detection that introduces a synergized adaptive noise scheduling mechanism. Experiments on several benchmark datasets demonstrate promising performance compared to prior diffusion-based anomaly detection methods.

**Strengths:**

1. The idea of customizing a synergized adaptive noise scheduling mechanism within a diffusion model is innovative and potentially valuable for anomaly detection.

2. The introduction of a semi-deterministic generative process is a meaningful adaptation. It helps reduce randomness during reconstruction and could indeed be beneficial for stable anomaly detection tasks.

**Weaknesses:**

1. Questionable learning effectiveness of deterministic diffusion. The paper modifies both the noise-adding and denoising processes from stochastic to deterministic forms. However, the theoretical rationale and empirical evidence supporting that such deterministic transitions preserve the learning capability of diffusion models are insufficient.

2. Unclear formulation of the SNR Scheduler. The proposed scheduler controls the noise coefficient α based on a target noise level M_T, yet the paper does not explain how M_T is derived, estimated, or learned.

3. Unaddressed risk of anomalous conditioning. The paper criticizes prior deterministic reconstruction methods for potentially conditioning on anomalous inputs but does not clearly explain how the proposed method avoids the same risk when performing semi-deterministic reconstruction.

 4. Ambiguous dimensionality description. During Data Preprocessing, the channel-wise concatenation operation changes the dimension from K to \hat{K}, but Equation (6) does not define \hat{K} or specify how concatenation is implemented among the original dimension K. This lack of clarity makes Equation (6) difficult to follow.

5. Questionable assumption. The proposed SNR Implicit Conditioning assumes that normal sequences exhibit lower frequencies while anomalous sequences exhibit higher frequencies. This assumption is not universally valid. Some anomalies manifest through amplitude or trend deviations rather than frequency differences. Moreover, for frequency-dominant anomalies, existing frequency-domain methods could already achieve superior performance, diminishing the uniqueness of the proposed mechanism.

6. The proof in Section 7.1 lacks clarity. The logical connection among Equations (19), (20), and (21) is difficult to follow, and the derivation steps between them are not well explained.

**Questions:**

See the weaknesses section

---

> ### Author Response · Authors · 2025-11-20
>
> | ID | Concern | Reply |
> | --- | --- | --- |
> | W1 | Effectiveness of deterministic denoising | Our method is a *semi-deterministic* diffusion rather than deterministic generation (such as autoencoder). The noise injection process remains identical to that of standard diffusion models, while the denoising process is made semi-deterministic through the estimation of an optimal corruption factor and adaptive denoising steps. Unlike conventional denoising that starts from Gaussian noise, we partially corrupt the input as the starting point for denoising, thereby enabling semi-deterministic reconstruction of the input, without compromising the generative capability. Furthermore, the second row of the ablation studies in Table 2 (using only SIC) demonstrates that semi-deterministic diffusion leads to improved anomaly detection performance. |
> | W2 | Details of SNR Scheduler | The SNR scheduler is formally defined in Equation (9). It controls the generation of α through the ratio of $M_T$ to $M_0$. We do not set $M_T$ but a TSNR which directly define the worst SNR at the terminal step of the diffusion process. The SNR scheduler ensure that every step of the noise injection is controllable and measured by SNR. When compared to previous methods, such as DiffAD, many dataset is found to be corrupted entirely with only one step of noise injection, even with the suggested parameters provided by the authors. The SNR scheduler achieved two things: a gradual noise injection that preserved the structure of time series better, hence, a better denoising learning; and, remove the need of fine tuning three individual but interdependent parameters ($\beta_{min}$, $\beta_{max}$, $T$). Lastly, we show the ablation study for TSNR in Section 5.3 to show the effectiveness of our design. |
> | W3 | Risk of anomaly conditioning | First, previous diffusion-based methods do not achieve deterministic reconstruction of the input data. Specifically, they partition the input into a conditional part and a target reconstruction part, using the former as auxiliary information to guide the generation of the latter. This generation process remains highly stochastic, and the selection of the conditional portion may even rely on anomalous values, as discussed in Section 3.2. In contrast, our approach does not directly use the input data as a condition. Instead, it estimates the optimal corruption factor and denoising steps, which serve as implicit conditions to guide the diffusion model's generation, thereby avoiding such risks. |
> | W4 | Data preprocessing details | We have provided the pseudocode for data preprocessing in Appendix Section 7.2. |
> | W5 | Confusing “High Frequency” claim | Thank you for your feedback. We chose the words “high frequency as anomalies” as we wanted to express the abrupt change in normalcy as anomalies. However, in a windowed time series, anomalies might not be detected via high frequency. We will make the necessary amendments in the subsequent submission to explain this confusion. Furthermore, we would like to point out that the SIC’s core principle lies in selecting the optimal corruption factor and denoising steps based on the amount of information retained from the original signal during the noise-injection process, thereby enabling semi-deterministic reconstruction of the input data. By measuring the complexity of the input sequence, we quantify the extent of information degradation caused by noise injection. Intuitively, the more complex the input, the more noise injection is required, thereby preserving reconstruction fidelity. Therefore, SNR implicit conditioning does not rely on the assumption of high-frequency anomalies. |
> | W6 | Step by step proof Section 7.1 | We have show the derivation below. |
>
> *Proof of Section 7.1:* Regarding Equation (19), since $\sqrt{{\bar{\alpha}}_t}$ in the numerator is a constant and $Z_0$ is a variable, the expression can be simplified to:
> $$
> \begin{aligned}
> \mu^2\left(\sqrt{{\bar{\alpha}}_t}Z_0\right)+\sigma^2\left(\sqrt{{\bar{\alpha}}_t}Z_0\right)
> &= {\bar{\alpha}}_t\mu^2\left(Z_0\right)+{\bar{\alpha}}_t\sigma^2\left(Z_0\right) \\
> &= {\bar{\alpha}}_t\left(\mu^2\left(Z_0\right)+\sigma^2\left(Z_0\right)\right)
> \end{aligned}
> $$
> Similarly, the denominator can be simplified to:
> $$
> \mu^2\left(\sqrt{1-{\bar{\alpha}}_t}\epsilon_t\right)+\sigma^2\left(\sqrt{1-{\bar{\alpha}}_t}\epsilon_0\right)=(1-{\bar{\alpha}}_t)\left(\mu^2\left(\epsilon_0\right)+\sigma^2\left(\epsilon_0\right)\right)
> $$
> Since $\epsilon \sim N(0,I)$, we have $\mu^2\left(\epsilon_0\right)=0$, $\sigma^2\left(\epsilon_0\right)=1$
> Thus, Equation (19) can be simplified to:
> $$
> \text{SNR}_t=\frac{{\bar{\alpha}}_t(\mu^2\left(Z_0\right)+\sigma^2\left(Z_0\right))}{1-{\bar{\alpha}}_t}
> $$
> After obtaining Equation (20), the ISNR estimation value from the SNR implicit conditioning method is substituted into Equation (20) to inversely solve for the optimal corruption factor $\alpha^\ast$, as shown in Equation (21).

---

> > ### Comment · Reviewer_J9Mc · 2025-11-28
> >
> > I sincerely appreciate the clarifications and the effort invested in addressing my questions. Your responses have resolved the vast majority of my concerns.
> >
> > My final remaining concern is about the derivation of *Phase 1: Optimal Corruption Factor Estimation* in Eq. (13), which is the core contribution of the paper. Although Appendix 7.1 provides a proof for computing the optimal $\alpha^*$, I still have two questions:
> >
> > 1. **On the definition of SNR.**
> >    In the definition of SNR, the term $P(Z)$ is computed using Eq. (18). However, Eq. (18) does not seem to follow the standard definition of the signal-to-noise ratio. Could the authors provide references or prior works that support this specific formulation of $P(Z)$ and the resulting SNR definition?
> >
> > 2. **On substituting SNR with ISNR in Eq. (21).**
> >    The computation of $\bar{\alpha}$ in Eq. (20) is derived under the SNR definition given by Eqs. (17) and (18). However, Eq. (21) directly replaces SNR with ISNR, which is defined in Eq. (12). Is this substitution theoretically justified?
> >
> > I would be happy to raise my score if the authors could clarify these points.

---

> > > ### Author Response · Authors · 2025-11-28
> > >
> > > Thank you for your thoughtful review of our work and the effort invested in evaluating our submission.
> > >
> > > 1.In reference [1], for a discrete-time signal with period N, the power over one period is defined as follows:
> > > $$
> > > P = \frac{1}{N} \sum_{n=0}^{N-1} [x[n]]^2
> > > $$
> > > In engineering and simulation applications, this formula is also used to calculate the power of aperiodic finite-length discrete signals. This definition corresponds to the mean square value of the sequence, which equals the sum of the variance and the square of mean. Therefore, for time series data, the power can be expressed as Equation (18). The SNR is defined as the ratio between the power of the useful signal and that of the noise. In diffusion models, the useful signal is $\sqrt{\bar{\alpha}_t} Z_0$, and the noise component is $\sqrt{1-\bar{\alpha}_t}\,\epsilon_t$. Therefore, the SNR can be expressed as in Equation (17), from which Equation (20) follows directly through algebraic manipulation.
> > >
> > > [1]Proakis, J. G., and Manolakis, D. G. Digital Signal Processing: Principles, Algorithms, and Applications. 4th ed. Prentice Hall, 2007.
> > >
> > > 2.In diffusion models, the training process learns to recover from progressive noise-injection, which we measured through SNR. In Eq 21, We employ the ISNR to estimate the optimal corruption factor, which measures the amount of implicit conditional information that must be retained by assessing the simplicity of the input data. Thus, the two metrics convey equivalent physical interpretations. In particular, for intrinsically complex data, preserving only a small fraction of the original information is sufficient to guide the model toward input-consistent reconstruction, which corresponds to a lower signal-to-noise ratio and a lower simplicity score. Furthermore, the ablation results in Table 2, together with the denoising-step analysis in Appendix 7.4.5, provide empirical evidence supporting the validity of the proposed corruption-factor estimation.

---

### Official Review · Reviewer_2Pjm · 2025-11-01

**Soundness:** 3
**Presentation:** 3
**Contribution:** 3
**Rating:** 6
**Confidence:** 4

**Summary:**

This paper targets the challenge of diffusion-model TSAD, i.e., stochastic reconstructions that inflate false positives. It proposes ICDiffAD, built on two components: 1) SNR Scheduler re-parameterizes the noise schedule using a target SNR and a monotone corruption function, deriving $\alpha_t$ directly from the SNR goal. 2) SNR Implicit Conditioning decomposes each test window via a zero-phase Gaussian low-pass filter $G_{\sigma}$ to compute an ISNR, then converts ISNR into an optimal corruption factor, and snaps it to the nearest pretrained diffusion step. On five benchmarks (MSL, SMAP, SMD, PSM, SWaT), the proposed ICDiffAD improves average F1 versus prior diffusion models and stronger baselines. The ablation experiments show gains from both SNR Scheduler and SIC.

**Strengths:**

1) This paper crisply argues that stochastic reconstructions in diffusion models conflict with determinism requirement of TSAD, with intuitive illustrations and discussion. The motivation is interesting and clear.

2) The proposed method is simple yet effective, and its training control is physically interpretable.

2) The experiments and ablations are comprehensive, and the proposed method improves average F1 and reduces false positives in comparisons.

**Weaknesses:**

1) This paper appears to contain notation inconsistencies in SNR expression. Eq. (10) defines SNR(t) using  $\bar{\alpha}_T$ (terminal product) while the text underneath says “$\bar{\alpha}_t$ represents cumulative signal retention up to step $t$”, which can confuse readers about the intended instantaneous SNR.

2) This paper performs per-dataset TPE searches over filter size and reports the best F1 scores, implying supervised selection using test labels, which can overstate generalization. Clarifying a validation split or unsupervised criterion would help.

3) The discussion and analysis of diffusion-based TSAD methods is not comprehensive enough, and a comprehensive review of relevant papers in this field is needed.

4) Many applications care about event-level detection/latency. Adding event-level metrics would strengthen claims.

**Questions:**

1) This paper appears to contain notation inconsistencies in SNR expression. Eq. (10) defines SNR(t) using  $\bar{\alpha}_T$ (terminal product) while the text underneath says “$\bar{\alpha}_t$ represents cumulative signal retention up to step $t$”, which can confuse readers about the intended instantaneous SNR.

2) This paper performs per-dataset TPE searches over filter size and reports the best F1 scores, implying supervised selection using test labels, which can overstate generalization. Clarifying a validation split or unsupervised criterion would help.

3) The discussion and analysis of diffusion-based TSAD methods is not comprehensive enough, and a comprehensive review of relevant papers in this field is needed.

4) Many applications care about event-level detection/latency. Adding event-level metrics would strengthen claims.

---

> ### Author Response · Authors · 2025-11-20
>
> | ID | Concern | Reply |
> | --- | --- | --- |
> | W1,Q1 | Notation Inconsistency | Thank you for pointing this out. In Equation (10), $\bar\alpha_T$ should indeed be corrected to $\bar\alpha_t$. We will make this revision in the subsequent version. |
> | W2,Q2 | Overstate General | We conducted TPE search over different filter sizes primarily to investigate the impact of \sigma on model performance. After identifying the optimal parameters, we performed systematic ablation studies on \sigma and provided parameter configuration recommendations (see Lines 459–476). The results reported in the main table are based on the parameters derived from our ablation studies, not those obtained via TPE. Furthermore, since the training set in time series anomaly detection contains only normal samples, it is not feasible to partition a standard validation set for conventional evaluation. |
> | W3,Q3 | Additional discussion on Diffusion based TSAD | Thank you for your suggestion. We will conduct a comprehensive review of diffusion-based time series anomaly detection methods in the subsequent version of our paper. |
> | W4,Q4 | Additional Metrics Required | We will report the event-level detection results in Table R1.1 and include this result in the subsequent submission. Meanwhile, metrics based on affiliation and VUS serve as anomaly sequence-level evaluation criteria, providing references for detection prioritization. The corresponding experimental results can be found in Appendix Section 7.4.2. |
>
> Table R1.1: Point Adjustment F1 Score
>
> | Dataset |  | MSL |  |  | SMAP |  |  | SMD |  |  | PSM |  |  | SWaT |  |  | Average |
> | --- | --- | --- | --- | --- | --- | --- | --- | --- | --- | --- | --- | --- | --- | --- | --- | --- | --- |
> | Method |  | P | R | F1 | P | R | F1 | P | R | F1 | P | R | F1 | P | R | F1 | F1 |
> | Classical | IForst | 53.94 | 86.54 | 66.45 | 52.39 | 59.07 | 55.53 | 42.31 | 73.29 | 53.64 | 76.09 | 92.45 | 83.48 | 49.29 | 44.95 | 47.02 | 61.22 |
> |  | ECOD | 66.14 | 73.18 | 69.48 | 71.41 | 76.33 | 73.79 | 86.45 | 75.73 | 80.74 | 75.65 | 86.32 | 80.63 | 69.54 | 68.32 | 68.92 | 74.71 |
> |  | CBLOF | 59.38 | 69.91 | 64.22 | 80.31 | 71.39 | 75.59 | 79.70 | 76.98 | 78.32 | 83.01 | 90.96 | 86.80 | 71.35 | 80.29 | 75.56 | 76.10 |
> |  | PCA | 62.18 | 77.58 | 69.03 | 76.39 | 62.19 | 68.56 | 55.64 | 69.18 | 61.68 | 81.92 | 91.28 | 86.35 | 59.28 | 77.29 | 67.10 | 70.54 |
> | Representation | Deep SVDD | 91.92 | 76.63 | 83.58 | 89.93 | 56.02 | 69.04 | 78.54 | 79.67 | 79.10 | 95.41 | 86.49 | 90.73 | 80.42 | 84.45 | 82.39 | 80.97 |
> |  | DCDetector | 93.69 | 99.69 | 96.60 | 95.63 | 98.92 | 97.25 | 83.59 | 91.10 | 87.18 | 97.14 | 98.74 | 97.93 | 93.11 | 99.77 | 96.33 | 95.06 |
> | Recontruction | ATF-UAD | 90.18 | 94.34 | 92.21 | 92.82 | 96.49 | 94.62 | 88.29 | 92.37 | 90.28 | 95.93 | 94.18 | 95.05 | 93.19 | 97.28 | 95.19 | 93.47 |
> |  | AT | 92.09 | 95.15 | 93.59 | 94.13 | 99.40 | 96.69 | 89.40 | 95.45 | 92.33 | 96.91 | 98.90 | 97.89 | 91.55 | 96.73 | 94.07 | 94.92 |
> |  | FGANomaly | 87.58 | 90.29 | 88.91 | 93.34 | 90.27 | 91.78 | 85.63 | 89.27 | 87.41 | 95.34 | 97.52 | 96.42 | 88.74 | 90.98 | 89.85 | 90.87 |
> |  | TimesNet | 83.92 | 86.42 | 85.15 | 92.52 | 58.29 | 71.52 | 88.66 | 83.14 | 85.81 | 98.19 | 96.76 | 97.47 | 86.76 | 97.32 | 91.74 | 86.34 |
> |  | D3R | 88.19 | 87.09 | 87.64 | 86.37 | 89.09 | 87.71 | 87.74 | 96.49 | 91.91 | 93.84 | 99.21 | 96.45 | 83.09 | 83.00 | 83.04 | 89.35 |
> |  | SARAD | 91.28 | 93.59 | 92.42 | 90.28 | 94.13 | 92.16 | 88.14 | 87.39 | 87.76 | 95.37 | 95.27 | 95.32 | 92.46 | 96.33 | 94.36 | 92.40 |
> | Diffusion Model | DiffAD | 92.97 | 95.44 | 94.19 | 96.52 | 97.38 | 96.95 | 90.01 | 95.67 | 92.75 | 97.00 | 98.92 | 97.95 | 98.44 | 96.90 | 97.66 | 95.90 |
> |  | ImDiffusion | 89.30 | 86.38 | 87.82 | 87.71 | 96.18 | 91.75 | 95.20 | 95.09 | 95.14 | 98.11 | 97.53 | 97.82 | 89.88 | 84.65 | 87.19 | 91.94 |
> |  | Ours | 95.29 | 94.32 | 94.80 | 98.11 | 97.09 | 97.60 | 95.81 | 95.50 | 95.65 | 98.02 | 98.61 | 98.31 | 97.19 | 99.01 | 98.09 | 96.89 |

---

### Author Response · Authors · 2025-12-03
**Summary for AC Consideration(Part 1/2)**

Dear Area Chair,

We sincerely appreciate your time and effort in handling our submission. We tackle a fundamental limitation of diffusion-based time series anomaly detection (TSAD) methods. Their stochastic reconstruction process undermines consistency and reliability in TSAD. ICDiffAD bridges the gap between stochastic generation and input-consistent reconstruction of Diffusion-based methods for TSAD, yielding significant improvements over our baselines and achieving the best overall performance across five widely used benchmarks.

- **Core Contributions**
    - **SNR Scheduler for Controllable Noise Injection**

        **SNR Scheduler modulate noise injection** during training, replacing heuristic schedules (e.g., $\beta_{\min}$, $\beta_{\max}$, and $T$) with controllable SNR-defined noise. By progressively decaying SNR to preserve original signals, it enhances perception under varying corruption levels while **eliminating two heuristic parameters** ($\beta_{\min}$ and $\beta_{\max}$) for more interpretable control.

    - **SNR Implicit Conditioning (SIC) for input-consistent Reconstruction**

        **SIC determines optimal noise injection and denoising steps** from input sequence complexity. By reformulating TSAD denoising, it uses input information to guide **faithful and stable reconstructions** while significantly reducing inference time through early termination $(t \le T)$.

    - **Practical and Robust Diffusion-based TSAD**

        ICDiffAD **consistently outperforms** strong baselines (DiffAD, ImDiffusion) across all metrics and datasets (Tables 1,4), delivering significantly **lower inference latency** and **enhanced usability** through heuristic-free adaptive inference control.

    - **Comprehensive Evaluation on 5 TSAD Datasets**

        We validate ICDiffAD on **five widely-used TSAD benchmarks**, demonstrating **overall optimal performance across diverse datasets**, confirming both its effectiveness and real-world applicability.

- **Summary of Manuscript Updates**

    Based on the reviewers’ valuable feedback, we have updated the manuscript, with all revisions highlighted in blue.

    1. We expanded the survey of diffusion-based time series anomaly detection methods in the Related Work section.
    2. We corrected the notations in Equations (6) and (10), clarified the notations in Section 3.2, and specified that TSNR in Section 4.3 is user-defined.
    3. In Section 4.1, we restated the motivation of the SNR scheduler, emphasizing that our contribution lies in its adaptability to the input data.
    4. We replaced the term “semi-deterministic generation” with “input-consistent generation” and updated related terminology accordingly.
    5. We removed the assumption in Section 4.4 that “normal sequences have lower frequencies and anomalous sequences have higher frequencies,” and clarified that the SIC mechanism determines the optimal denoising step and corruption factor by adaptively assessing how much original signal information is preserved, independent of anomaly frequency.
    6. We added point-adjusted (event-level) evaluation in Appendix 7.4.2.
    7. Appendix 7.4.5 now includes an ablation study on fixed denoising steps.

---

> ### Author Response · Authors · 2025-12-03
> **Summary for AC Consideration(Part 2/2)**
>
> - **Summary of Rebuttal**
>
>     We summarize the main concerns and our responses below.
>
>     - **Clarifications:**
>
>
>         | Reviewer(s) | Concerns | Our Response | Reviewer concern addressed? |
>         | --- | --- | --- | --- |
>         | 2Pjm | overestimation of generation performance by TPE | TPE search aims to investigate the effect of σ on model performance and to provide guidelines for parameter configuration (Lines 459–476). The performance evaluation in the manuscript is based on parameters obtained from the ablation study, rather than from TPE. | no reply |
>         | J9Mc, SPCE | the term ‘semi-deterministic generation’ in diffusion models | Following reviewer SPCE’s suggestion, we adopted the term ‘input consistency’ in the revised manuscript to replace ‘semi-deterministic,’ thereby avoiding potential misunderstandings. | Yes |
>         | J9Mc, SPCE | SNR scheduler usage details and motivation | In Section 4.3 of the revised manuscript, we explicitly state that TSNR is user-specified to clarify the usage details of the SNR scheduler. Moreover, following reviewer SPCE’s suggestion, we revised Section 4.1 to restate the motivation of SNR and highlight that our contribution lies in its adaptability to the input data. | Yes |
>         | J9Mc | risk of anomalous conditioning | The diffusion-model baselines use partial observations as auxiliary information to guide the generation of unobserved values, which may include anomalous segments, as noted in Section 3.2. In contrast, our method estimates the optimal noise factor and denoising steps as implicit conditions for guiding the diffusion process, thereby avoiding such risks. | Yes |
>         | J9Mc, uRQA | the assumption of high-frequency anomaly patterns | Our description of ‘high frequency as anomalies’ led to a misunderstanding. We have revised this wording in the updated manuscript and clarified that the core principle of SIC lies in selecting the optimal noise level and denoising steps as implicit conditions based on the amount of information preserved from the original signal during noise injection. The mechanism does not rely on the assumption of high-frequency anomalies. | Yes |
>         | J9Mc | Derivation and rationale of optimal corruption factor estimation | In our response to reviewer J9Mc, we provided details of the formula derivation, cited references to support the definition of SNR, and explained the rationale behind the ISNR estimation. | no reply |
>         | SPCE, uRQA | Inference speed faster than diffusion model baselines | Although our method has higher GFLOPS than the diffusion model baseline, due to the latter’s overhead from masked generation and our method’s adaptive estimation of the optimal denoising factor and number of denoising steps, we achieve faster inference speed. | Yes |
>     - **New experiments and metric:**
>
>
>         | Reviewer(s) | Concerns | Our Response | Reviewer concern addressed? |
>         | --- | --- | --- | --- |
>         | 2Pjm, SPCE | Additional Metrics Required | We evaluated our method using point-adjusted (event-level) metrics, and the results have been added to Appendix 7.4.2. | Yes |
>         | SPCE | Effectiveness of SIC | We conducted experiments using a fixed set of reconstruction starting steps, and these results have been added to Appendix 7.4.5. | Yes |
> - **Rebuttal Updates**
>     - Reviewer SPCE has already raised score and indicated that they are willing to raise it further after issues W1 and W4 are addressed in the revised manuscript. Reviewer J9Mc expressed willingness to raise score after we clarified the derivation of Equation (13). Although they were subsequently unable to continue due to a bug, we believe that our responses and revisions adequately address their concerns.
>     - Although reviewer 2Pjm did not participate in the rebuttal, all of their comments were constructive suggestions or requests for clarification, and we have addressed each of them in our responses and revisions.
>     - Reviewer uRQA has already raised score, and in the revised manuscript we have made corresponding revisions to the statements they were concerned about, such as the introduction of SNR and how it benefits time series.
>
> Thank you again for your consideration.
>
> Best regards
>
> Authors of Paper 5923

---

### Meta-Review · Area_Chair_pBeg · 2026-01-05

**Summary:**

This paper considers a failure mode of diffusion-model based time-series anomaly detection - namely, the inconsistencies introduced by stochastic reconstructions. Towards this end, the authors propose an SNR-guided scheduler and an implicit conditioning approach to enable input-consistent reconstructions.

The overall motivation is sound, and the problem that is being addressed is interesting. The evaluation with five datasets makes this more compelling.

 Reviewers were satisfied with some of the revisions/justifications provided, but some concerns remained, including the current framework / solution to not be fully convincing and concerns about the methodology (e.g. the benefit of the nature of the \alpha curve).  There is also a lingering concern pertaining to the corruption factor setup and the ISNR component.

**Reviewer Concerns:**

Reviewers were satisfied with some of the revisions/justifications provided, but some concerns remained, including the current framework / solution to not be fully convincing and concerns about the methodology (e.g. the benefit of the nature of the \alpha curve).  There is also a lingering concern pertaining to the corruption factor setup and the ISNR component.

**Reviewer Scores:**

Had there been a full discussion, after looking at the changes and the rebuttal, I conjecture that the reviewer scores would have gone up by a point.

---

### Decision · Program_Chairs · 2026-01-26

Accept (Poster)